# Evolution of gene dosage on the Z-chromosome of schistosome parasites

Marion A L Picard[1], Celine Cosseau[2], Sabrina Ferré[3], Thomas Quack[4], Christoph G Grevelding[4], Yohann Couté[3], Beatriz Vicoso[1]*

[1]Institute of Science and Technology Austria, Klosterneuburg, Austria; [2]University of Perpignan Via Domitia, IHPE UMR 5244, CNRS, IFREMER, University Montpellier, Perpignan, France; [3]Université Grenoble Alpes, CEA, Inserm, BIG-BGE, Grenoble, France; [4]Institute for Parasitology, Biomedical Research Center Seltersberg, Justus-Liebig-University, Giessen, Germany

**Abstract** XY systems usually show chromosome-wide compensation of X-linked genes, while in many ZW systems, compensation is restricted to a minority of dosage-sensitive genes. Why such differences arose is still unclear. Here, we combine comparative genomics, transcriptomics and proteomics to obtain a complete overview of the evolution of gene dosage on the Z-chromosome of *Schistosoma* parasites. We compare the Z-chromosome gene content of African (*Schistosoma mansoni* and *S. haematobium*) and Asian (*S. japonicum*) schistosomes and describe lineage-specific evolutionary strata. We use these to assess gene expression evolution following sex-linkage. The resulting patterns suggest a reduction in expression of Z-linked genes in females, combined with upregulation of the Z in both sexes, in line with the first step of Ohno's classic model of dosage compensation evolution. Quantitative proteomics suggest that post-transcriptional mechanisms do not play a major role in balancing the expression of Z-linked genes.
DOI: https://doi.org/10.7554/eLife.35684.001

**\*For correspondence:**
beatriz.vicoso@ist.ac.at

**Competing interests:** The authors declare that no competing interests exist.

## Introduction

In species with separate sexes, genetic sex determination is often present in the form of differentiated sex chromosomes (*Bachtrog et al., 2014*). A sex-specific chromosome can be carried by the male (such as the Y of mammals and fruit flies, in male heterogamety) or by the female (such as the W of birds, in female heterogamety). These sex chromosomes originally arise from pairs of autosomes, which stop recombining after they acquire a sex-determining region (*Charlesworth, 1991*; *Charlesworth et al., 2005*). The loss of recombination between X/Z and Y/W chromosomes is likely driven by selective pressures to link the sex-determining gene and alleles with sexually antagonistic effects, and often occurs through inversions on the sex-specific chromosome (*Rice, 1987*; *Bergero and Charlesworth, 2009*). The inverted Y/W-linked region stops recombining entirely, which hampers the efficacy of selection and leads to its genetic degeneration (*Charlesworth et al., 2005*; *Engelstädter, 2008*). The appearance of further sexually antagonistic mutations can restart the process and select for new non-recombining regions, creating sex chromosome 'strata' of different ages (*Ellegren, 2011*; *Wang et al., 2012*; *Vicoso et al., 2013a*; *Vicoso et al., 2013b*). Eventually, this suppression of recombination can extend to most of the chromosome, leading to gene-poor, mostly heterochromatic sex chromosomes such as the Y chromosome of mammals (*Lemaitre et al., 2009*).

The loss of one gene copy on the Y/W is predicted to result in a two-fold reduction of expression in the heterogametic sex, as gene expression is correlated with gene copy number (*Guo et al., 1996*). This can cause imbalances in gene networks composed of both X/Z-linked and autosomal genes (*Wijchers and Festenstein, 2011*). Such imbalances can drive the appearance of dosage

**eLife digest** The DNA inside cells is organized in structures called chromosomes, some of which can control whether individuals develop as males or females. For instance, female mammals have two X chromosomes, whereas male mammals have one X and one Y chromosome. A mechanism called 'dosage compensation' makes sure that females do not produce double the number of transcripts from genes on the X-chromosome as males.

In other organisms, including the parasitic flatworms called Schistosomes, females have ZW sex chromosomes, whereas males have two Z chromosomes. In these parasites, males do create more transcripts from genes on the Z chromosome than females do, suggesting they do not have the same kind of compensation mechanisms as mammals.

Among Schistosome parasites, the Z chromosome has only been studied in detail in the model organism *Schistosoma mansoni*. Investigating other closely related species can shed light on how the Z and W chromosomes evolved.

Picard et al. studied the Z chromosome in two additional species of Schistosome parasites: the African *S. haematobium* and the Asian *S. japonicum*. Using a technique called DNA sequencing, Picard et al. were able to analyse their genes, focusing on a part of the Z chromosome known to have been lost from the W chromosome. The results revealed that this region was different in the African and Asian species. In addition, females of both species expressed genes on their single Z chromosome at fairly high levels. The males did not need to express these genes at a high level because they have two copies – but they did so anyway. This could be because this high expression is a by-product of the way the females have evolved to boost their Z chromosome gene expression.

A next step will be to investigate the molecular mechanisms underlying this regulation. Schistosomiasis – a disease caused by this type of flatworm parasite – is one of the deadliest neglected tropical diseases, according to the US Centers of Disease Control. It kills more than 200,000 people a year. Better understanding of the reproductive biology of this parasite could eventually help to develop ways to control it by interfering with its reproduction.

DOI: https://doi.org/10.7554/eLife.35684.002

compensation mechanisms, which target X/Z chromosomes and regulate their expression to restore optimal dosage (*Ohno, 1967*; *Gartler, 2014*). While X/Z upregulation in the heterogametic sex is required to re-establish balanced levels of expression, global downregulation in the homogametic sex is also observed (e.g. X-inactivation in mammals). Ohno suggested a two-step mechanism, in which the initial upregulation of expression is not sex-specific. This leads to an excess of dosage in the homogametic sex, and secondarily selects for further repressing mechanisms ('Ohno's hypothesis' of dosage compensation *Ohno, 1967*]). How relevant this model is to the evolution of mammalian dosage compensation is still under debate (e.g. *Gu and Walters, 2017*; *Lin et al., 2012*; *Nguyen and Disteche, 2006*; *Vicoso and Bachtrog, 2009a*; *Mank, 2013*). Independent of the underlying mechanisms, balanced gene expression between males and females in species with differentiated sex chromosomes was used as diagnostic of a chromosome-wide (also referred to as 'global', or 'complete') mechanism of dosage compensation in many different clades (*Vicoso and Bachtrog, 2009a*; *Mank, 2013*; *Gu and Walters, 2017*).

In ZW systems, the loss of genes on the sex-specific W chromosome is generally accompanied by unequal expression levels of the Z-chromosome between ZZ males and ZW females, as well as reduced expression of the Z relative to the autosomes in females. This has generally been interpreted as a lack of chromosome-wide dosage compensation (also referred to as 'partial', or 'incomplete'), with individual dosage-sensitive genes being independently regulated instead. Incomplete dosage compensation was described in a wide range of species, including birds (*Itoh et al., 2007*; *Ellegren et al., 2007*; *Arnold et al., 2008*; *Wolf and Bryk, 2011*), fishes (*Chen et al., 2014*) and snakes (*Vicoso et al., 2013a*). So far, Lepidoptera are the only exception to this observation (*Gu and Walters, 2017*; *Huylmans et al., 2017*). Why many ZW systems should fail to acquire a global mechanism of dosage compensation is not entirely clear, although several and non-mutually exclusive hypotheses have been put forward (see Discussion, and *Gu and Walters [2017]* for a review). Another possibility is that the male-bias of the Z is instead caused by an accumulation of

genes with male functions due to the male-biased transmission of the Z, which may favor the fixation of sexually antagonistic male-beneficial mutations on this chromosome.

While the direct comparison of male and female expression of X/Z-linked and autosomal genes has provided an overview of dosage compensation in many clades, it suffers from several drawbacks (*Gu and Walters, 2017*). First, chromosome-wide dosage compensation can lead to strongly sex-biased expression, if only the initial upregulation of expression in both sexes has occurred (but not the secondary downregulation of Ohno's hypothesis [*Ohno, 1967*]). This has been suggested for the flour beetle (*Prince et al., 2010*) and for the young sex chromosomes of the threespine stickleback (*Schultheiß et al., 2015*). Biases in expression levels between sexes and/or chromosomes may also have been present ancestrally, before the present sex-chromosomes evolved, and using a proxy for ancestral expression can yield insights into the direct consequences of sex-linkage (*Julien et al., 2012*; *Vicoso and Bachtrog, 2015*; *Gu et al., 2017*). Finally, the vast majority of studies relied only on microarray or RNA-seq data and did not consider any post-transcriptional regulation that might affect gene dosage at the protein level, but not at the transcript level (whereas protein dosage is in most cases the functionally relevant measure). For instance, a proteomic analysis in birds found that several genes appeared to be partially equalized at the protein level despite being strongly male-biased at the transcript level (*Uebbing et al., 2015*). In humans, post-transcriptional regulation does not appear to play a major role in dosage compensation (*Chen and Zhang, 2015*).

Here, we combine comparative genomics, transcriptomics and quantitative proteomics to obtain a complete overview of the evolution of gene dosage on the Z-chromosome of parasites of the genus *Schistosoma*. Schistosomes are a group of blood parasites that can cause schistosomiasis in humans (*Chitsulo et al., 2004*). Their complex life cycle is characterized by a phase of clonal multiplication in an intermediate mollusk host, and a phase of sexual reproduction in the final warm-blooded host. Unlike the other 20,000 species of hermaphroditic platyhelminths, schistosomes have separate sexes: sexual reproduction occurs immediately after the primary development of males and females in their definitive host, and mating is compulsory for the sexual maturation of females (*Loker and Brant, 2006*; *Kunz, 2001*). Sex determination is genetic, and relies on a pair of cytogenetically well-differentiated ZW chromosomes (*Grossman et al., 1981a*). All schistosomes are thought to share the same ancestral pair of ZW sex chromosomes, but differences in their morphology and in the extent of heterochromatization of the W suggest that different strata were acquired independently by different lineages (*Grossman et al., 1981a*; *Lawton et al., 2011*).

The model blood fluke *Schistosoma mansoni* was one of the first ZW clades to be evaluated for the presence of global dosage compensation, through the comparison of male and female microarray data derived from several tissues (*Vicoso and Bachtrog, 2011*). It showed reduced expression of Z-linked genes in females relative (i) to the autosomes and (ii) to males, consistent with a lack of chromosome-wide dosage compensation. Interestingly, the reduction of Z-expression in females was less than two-fold, and the Z:autosome ratio of expression was slightly, but consistently, greater than one in males. Our combined genomic, transcriptomic, and proteomic approaches allow us to fully probe the evolution of the male-biased expression of the Z, and suggest a more complex scenario than previously proposed. We discuss this in light of the different hypotheses put forward to account for the evolution of gene dosage on Z chromosomes.

## Results

### Genomic differentiation of ZW sex chromosomes in Asian and African lineages

The difference in morphology of the ZW pair in African and Asian schistosomes suggests that the two lineages may differ in their gene content (*Grossman et al., 1981a*). We compared the gene content of the Z-chromosomes of three different species: *S. mansoni* and *Schistosoma haematobium*, which belong to African schistosomes, and *Schistosoma japonicum*, an Asian schistosome (*Figure 1*). We first identified syntenic blocks between the *S. mansoni* genome and the *S. haematobium* and *S. japonicum* scaffolds. To this end, we mapped all *S. mansoni* protein coding sequences to the genome assemblies of the two other species and selected only the hits with the highest scores, yielding 9504 *S. mansoni*/*S. haematobium* orthologs and 8555 *S. mansoni*/*S. japonicum* orthologs

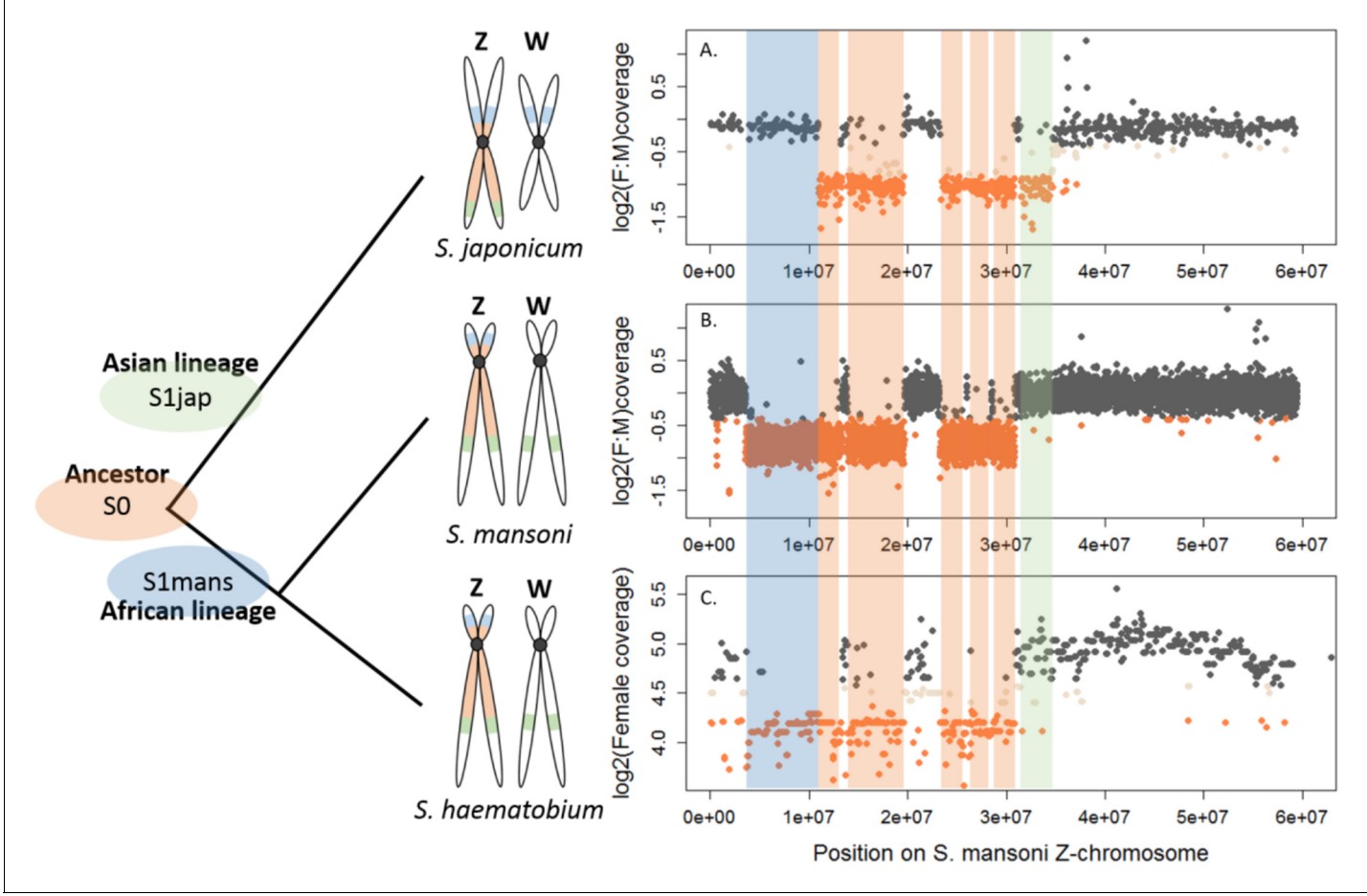

**Figure 1.** Shared and lineage-specific evolutionary strata on the Z-chromosome. The phylogeny of the three species is represented on the left. The female:male (F:M) ratio of coverage (y-axis) along the Z-chromosome of *S. mansoni* (x-axis) is shown for *S. japonicum* scaffolds (**A**) and *S. mansoni* 10 kb windows (**B**). Female coverage is shown for *S. haematobium* scaffolds (**C**). All species share an ancestral Z-linked stratum S0 (marked in orange). The stratum S1jap (in green) is specific to the Asian lineage represented by *S. japonicum*. The stratum S1mans (in blue) is specific to the African lineage, represented by *S. mansoni* and *S. haematobium*. Dot color is attributed depending on the window/scaffold location within each species: Z-specific regions in orange, pseudoautosomal regions in grey, and ambiguous regions in beige.

DOI: https://doi.org/10.7554/eLife.35684.003

The following source data is available for figure 1:

**Source data 1.** Comparative genomics: coverage analysis and strata identification.
DOI: https://doi.org/10.7554/eLife.35684.004

**Source data 2.** *S. mansoni* reference species: coverage analysis and Z/Autosome assignment.
DOI: https://doi.org/10.7554/eLife.35684.005

**Source data 3.** *S. haematobium* autosome vs Z-specific region assigment.
DOI: https://doi.org/10.7554/eLife.35684.006

**Source data 4.** *S. japonicum* autosome vs Z-specific region assignment.
DOI: https://doi.org/10.7554/eLife.35684.007

(*Table 1*). Scaffolds were then assigned to one of the *S. mansoni* chromosomes, based on their ortholog content (*Figure 1—source data 1*).

We further performed a comparative coverage analysis to define the Z-specific regions of the three species. Z-derived sequences are expected to display half the genomic coverage in ZW females as in ZZ males, and as the autosomes. We thus mapped male and female genomic reads (or only female reads in the case of *S. haematobium*) to the reference genome of each species (*Protasio et al., 2012a*; *Criscione et al., 2009*; *Young et al., 2012*; *Zhou et al., 2009*). Publicly available raw reads were used for *S. mansoni* and *S. haematobium* (Wellcome Trust Sanger Institute

**Table 1.** Number of orthologs assigned as Z-linked and autosomal in *S. mansoni*, *S. haematobium* and *S. japonicum*, based on the female:male (or female for *S. haematobium*) coverage patterns.

| | Categories | Schistosoma japonicum | | | | Schistosoma haematobium | | | |
|---|---|---|---|---|---|---|---|---|---|
| | | Z-specific | Autosomal | Ambiguous | Not covered | Z-specific | Autosomal | Ambiguous | Not covered |
| *Schistosoma mansoni* | Z-specific | 476 (*S0*) | 306 (*S1mans*) | 20 | 3 | 847 | 36 | 23 | 10 |
| | Autosomal | 137 (*S1jap*) | 7062 | 91 | 13 | 216 | 7462 | 262 | 105 |
| | Excluded | 57 | 383 | 4 | 3 | 105 | 411 | 20 | 7 |
| | Orthologs total | 8555 | | | | 9504 | | | |

DOI: https://doi.org/10.7554/eLife.35684.008

Bioprojects PREJB2320 and PREJB2425), whereas male and female *S. japonicum* were sequenced for this study. We then estimated the per base genomic coverage. Median coverage values were 18.40 and 18.99 for *S. mansoni* male and female libraries; 23.5 and 7.43 for *S. haematobium* female#1 and female#2 libraries; 23.77 and 20.53 for *S. japonicum* male and female libraries. Z-specific genomic regions were defined by a maximum value of the female:male ratio of coverage (*S. mansoni*: log2(female:male)=−0.4; *S. japonicum*: log2(female:male)=−0.84), or a maximum value of female coverage (*S. haematobium*: log2(female)=4.41). Details of how these cutoff values were obtained are provided in the Materials and methods and Appendix 1. This analysis resulted in 285 newly described Z-specific genes in *S. mansoni* that were previously located on 19 unplaced scaffolds longer than 50 kb, and to a refined pseudoautosomal/Z-specific structure of the published ZW linkage group (*Protasio et al., 2012a*; *Criscione et al., 2009*) (*Figure 1—source data 2*). It further allowed us to define 379 Z-specific scaffolds (containing 1409 annotated genes with orthologs in *S. mansoni*) in *S. haematobium* (*Figure 1—source data 3* for exhaustive list) and 461 Z-specific scaffolds (containing 706 annotated orthologs) in *S. japonicum* (*Figure 1—source data 4* for exhaustive list).

While the content of the Z was largely shared between the African *S. mansoni* and *S. haematobium* (*Table 1*, *Figure 1*), large differences were found between the African and Asian lineages: only 476 Z-specific genes were shared by *S. mansoni* and *S. japonicum*, while 306 were only Z-specific in *S. mansoni* and 137 only in *S. japonicum* (*Table 1*). Of all these Z-specific genes, 613 were already mapped to the *S. mansoni* ZW linkage group (*Protasio et al., 2012a*; *Criscione et al., 2009*) and, when plotted along the Z-chromosome, outlined three different evolutionary strata: one shared ancestral stratum (S0: 367 genes) and two lineage-specific strata (S1mans, specific to the African schistosomes, with 180 genes; and S1jap, specific to *S. japonicum*, with 66 genes) (*Table 1*, *Figure 1*, and *Figure 1—source data 1*). The presence of pseudoautosomal regions throughout the S0 (*Figure 1*) is likely due to errors in the genome assembly. All further analyses were run using all newly identified Z-specific genes, but hold when only Z-specific genes that were previously mapped to the ZW linkage group are considered (Appendix 1).

## Consistent patterns of expression in *S. mansoni* and *S. japonicum*

In order to test for dosage compensation, the median expression of Z-specific genes in ZW females can be compared to the median autosomal expression (Z:AA ratio) and/or to the Z-specific gene expression in ZZ males (F:M ratio). Z:AA or F:M ratio of ~1 supports global dosage compensation, while a ratio between 0.5 and 1 suggests partial or local dosage compensation. We performed this analysis in *S. mansoni* and in *S. japonicum*, using publicly available RNA-seq reads derived from a sexually undifferentiated stage (schistosomula, [*Picard et al., 2016*; *Wang et al., 2017*]) and a sexually mature stage (adults, Wellcome Trust Sanger Institute Bioproject PRJEB1237, [*Wang et al., 2017*]). The inclusion of a sexually undifferentiated stage (which lack primary spermatocytes or eggs) is important, as much of the expression obtained from adults will necessarily come from their well-developed gonads. Sex-linked genes are often sex-biased in the germline, even in organisms that have chromosome-wide dosage compensation (e.g. due to sex-chromosome inactivation during gametogenesis), and the inclusion of gonad expression has led to inconsistent assessments of the status of dosage compensation in other clades (*Gu and Walters, 2017*; *Huylmans et al., 2017*).

Reads were mapped to their respective genomes, and expression values in Reads Per Kilobase Million (RPKM) were calculated for each gene (*Figure 2*, *Figure 2—source data 1* and *2*); only genes with a minimum RPKM of 1 in both sexes were considered.

We consistently observed a strong male bias in the expression of Z-specific genes in both stages and species (F:M ratio between 0.58 and 0.69, *Figure 2* and *Supplementary file 1*), consistent with local or incomplete dosage compensation. While this was generally supported by the lower expression levels of Z-specific genes in females when compared to the autosomes (Z:AA ratio between 0.73 and 0.85; *Figure 2*, *Supplementary file 1*), this difference was only apparent for some filtering procedures (*Figure 2—figure supplement 1* to *Figure 2—figure supplement 14*), and even then was not sufficient to fully account for the strong male-bias of the Z. Instead, the higher expression of the male Z in both stages and species (ZZ:AA ratio between 1.25 and 1.46, *Supplementary file 1*) appeared to also contribute to the male-bias of Z-linked genes. These patterns were qualitatively robust to changes in the methods used to estimate expression (RPKM or TPM [Transcripts Per Kilobase Million]), in the filtering procedure (RPKM > 0, RPKM > 1, TPM >0 or TPM >1), and when only genes that were previously mapped to the ZW linkage group were considered. These analyses were further performed independently in the S0, S1mans and S1jap strata, which showed no significant difference in the extent of their male bias. All the resulting plots are shown in *Figure 2—figure supplement 1* to *Figure 2—figure supplement 14*. Finally, Z-specific genes were found to be male-biased even when only genes with broad expression were considered (RPKM > 1 and RPKM > 3 in all samples, and when genes with strong sex-biases in expression were excluded (M:F > 2 or F: M > 2, *Figure 2—figure supplement 15* and *Figure 2—figure supplement 16, I–L* panels), confirming that this pattern does not appear to be driven simply by the presence of genes with sex-specific functions on the Z-chromosome. No further influence of known protein-protein interactions was detected (*Figure 2—figure supplement 17*, Appendix 1).

## Convergent upregulation of the Z in both sexes

The previous patterns are consistent with an upregulation of the Z-chromosome in both sexes after the degeneration of the W-specific region, and could represent the intermediate step in the evolution of dosage compensation originally postulated by Ohno. However, they could also be due to high expression of the ancestral proto-Z in both sexes, before sex chromosome divergence. To exclude this, we identified one-to-one orthologs between genes annotated in both species using a reciprocal best hit approach (7382 orthologs, *Figure 3—source data 1*). All genes that were classified as Z-specific in one species but as autosomal in the other were considered to be part of the S1 strata (S1jap if they were Z-specific in *S. japonicum* or S1mans if they were Z-specific in *S. mansoni*). We then used the pseudoautosomal expression of these lineage-specific strata as a proxy for the ancestral level of expression. For instance, in *S. japonicum*, we estimated the S1jap:AA ratio, after normalizing the expression data by their respective (pseudo)autosomal level in *S. mansoni* (*Figure 3A and B*). The reversed analysis was performed for S1mans (*Figure 3C and D*).

*Figure 3* confirms that the male-biased expression of Z-specific genes is a consequence of their sex-linkage, and that the Z-chromosome has become under-expressed in females relative to the ancestral expression. However, a full two-fold reduction in female expression is not observed, consistent with partial upregulation, and/or full upregulation of a subset of dosage-sensitive genes (Z:AA ranging from 0.68 to 0.83, *Supplementary file 1*)(*Sangrithi et al., 2017*; *Pessia et al., 2012*). *Figure 3* also generally supports an increase in expression in males (ZZ:AA ranging from 0.98 to 1.35, *Supplementary file 1*). Male adults of *S. japonicum* are the exception, with a ZZ:AA of 0.98. However, given that an excess of expression is observed when (**i**) we do not take into account the ancestral expression (*Figure 2*), (ii) we focus on genes previously mapped to the ZW pair (*Figure 2—figure supplements 3*, *4*, *10* and *11*), and (iiii) we consider the schistosomula stage (with or without the ancestral expression and independent of the classification), this is likely due to noise in the sample and not to a true biological difference (only 58 genes were tested). *Figure 3* shows the distributions for all genes with a minimum RPKM value of 1 in males and females of both species. We repeated the analysis using the same filters as before (minimum RPKM of 0, TPM of 0, TPM of 1), and with a publicly available list of 1:1 orthologs (obtained from the Wormbase Biomart, see Methods). The resulting plots are shown in *Figure 2—figure supplement 1* to *Figure 2—figure supplement 14*, and *Figure 3—figure supplements 1* and *2*. Gene expression values for the orthologs of each species are provided in *Figure 3—source data 2* and *3*.

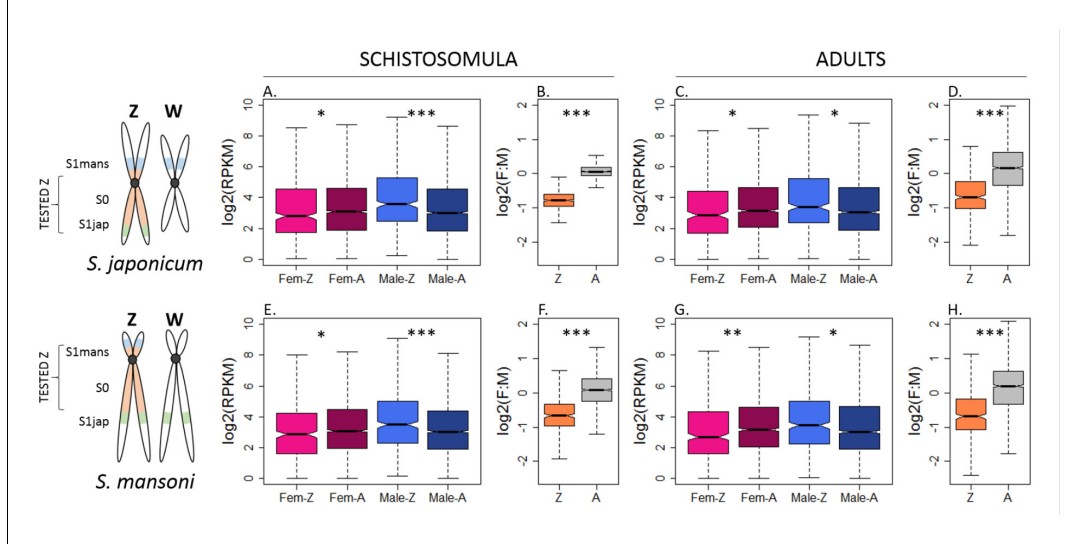

**Figure 2.** Patterns of expression on the Z and autosomes of *S.japonicum* and *S. mansoni*. Z-linked and autosomal gene expression patterns are shown for *S. japonicum* (**A-D**) and *S. mansoni* (**E-H**), in undifferentiated schistosomula and sexually mature adults. In panels A, C, E, and G, Fem-Z and Male-Z refer to the expression of Z-linked genes in females and males, respectively, and Fem-A and Male-A to the expression of the autosomal genes in females and males. In panels B, D, F, and H, Z refers to Z-linked genes and A to autosomal genes. In all conditions, a strong male bias is observed for the Z-linked genes (**B, D, F, H**). This male-biased expression of the Z-linked genes is accompanied by both an under-expression in females and an over-expression in males, compared to the level of autosomal expression (**A, C, E, G**). The level of significance of each comparison (Wilcoxon rank sum test with continuity correction) is indicated by asterisks: *p-value<0.05, **p-value<0.001, ***p-value<0.0001.

DOI: https://doi.org/10.7554/eLife.35684.009

The following source data and figure supplements are available for figure 2:

**Source data 1.** RPKM calculation for *S. mansoni* gene expression.

DOI: https://doi.org/10.7554/eLife.35684.027

**Source data 2.** RPKM calculation for *S. japonicum* gene expression.

DOI: https://doi.org/10.7554/eLife.35684.028

**Figure supplement 1.** Adult expression patterns (RPKM>1, exhaustive strata) of genes located in the different strata of the Z (S0, S1man and S1jap), as well as pseudoautosomal and autosomal genes.

DOI: https://doi.org/10.7554/eLife.35684.010

**Figure supplement 2.** Adult expression patterns (RPKM>0, exhaustive strata) of genes located in the different strata of the Z (S0, S1man and S1jap), as well as pseudoautosomal and autosomal genes.

DOI: https://doi.org/10.7554/eLife.35684.011

**Figure supplement 3.** Adult expression patterns (RPKM>1, stringent strata) of genes located in the different strata of the Z (S0, S1man and S1jap), as well as pseudoautosomal and autosomal genes.

DOI: https://doi.org/10.7554/eLife.35684.012

**Figure supplement 4.** Adult expression patterns (RPKM>0, stringent strata) of genes located in the different strata of the Z (S0, S1man and S1jap), as well as pseudoautosomal and autosomal genes.

DOI: https://doi.org/10.7554/eLife.35684.013

**Figure supplement 5.** Adult expression patterns (RPKM>1, exhaustive strata, Wormbase orthologs) of genes located in the different strata of the Z (S0, S1man and S1jap), as well as pseudoautosomal and autosomal genes.

DOI: https://doi.org/10.7554/eLife.35684.014

**Figure supplement 6.** Adult expression patterns (TPM>1, exhaustive strata) of genes located in the different strata of the Z (S0, S1man and S1jap), as well as pseudoautosomal and autosomal genes.

DOI: https://doi.org/10.7554/eLife.35684.015

**Figure supplement 7.** Adult expression patterns (TPM>0, exhaustive strata) of genes located in the different strata of the Z (S0, S1man and S1jap), as well as pseudoautosomal and autosomal genes.

DOI: https://doi.org/10.7554/eLife.35684.016

**Figure supplement 8.** Schistosomula expression patterns (RPKM>1, exhaustive strata) of genes located in the different strata of the Z (S0, S1man and S1jap), as well as pseudoautosomal and autosomal genes.

DOI: https://doi.org/10.7554/eLife.35684.017

**Figure supplement 9.** Schistosomula expression patterns (RPKM>0, exhaustive strata) of genes located in the different strata of the Z (S0, S1man and S1jap), as well as pseudoautosomal and autosomal genes.

*Figure 2 continued*

DOI: https://doi.org/10.7554/eLife.35684.018

**Figure supplement 10.** Schistosomula expression patterns (RPKM>1, stringent strata) of genes located in the different strata of the Z (S0, S1man and S1jap), as well as pseudoautosomal and autosomal genes.

DOI: https://doi.org/10.7554/eLife.35684.019

**Figure supplement 11.** Schistosomula expression patterns (RPKM>0, stringent strata) of genes located in the different strata of the Z (S0, S1man and S1jap), as well as pseudoautosomal and autosomal genes.

DOI: https://doi.org/10.7554/eLife.35684.020

**Figure supplement 12.** Schistosomula expression patterns (RPKM>1, exhaustive strata, Wormbase orthologs) of genes located in the different strata of the Z (S0, S1man and S1jap), as well as pseudoautosomal and autosomal genes.

DOI: https://doi.org/10.7554/eLife.35684.021

**Figure supplement 13.** Schistosomula expression patterns (TPM>1, exhaustive strata) of genes located in the different strata of the Z (S0, S1man and S1jap), as well as pseudoautosomal and autosomal genes.

DOI: https://doi.org/10.7554/eLife.35684.022

**Figure supplement 14.** Schistosomula expression patterns (TPM>0, exhaustive strata) of genes located in the different strata of the Z (S0, S1man and S1jap), as well as pseudoautosomal and autosomal genes.

DOI: https://doi.org/10.7554/eLife.35684.023

**Figure supplement 15.** Z-linked and autosomal female:male ratio of gene expression using different filters.

DOI: https://doi.org/10.7554/eLife.35684.024

**Figure supplement 16.** Z-linked and autosomal gene expression in females and males using different filters.

DOI: https://doi.org/10.7554/eLife.35684.025

**Figure supplement 17.** Z-linked and autosomal female:male ratio of gene expression according to presence/absence of known protein-protein interactions.

DOI: https://doi.org/10.7554/eLife.35684.026

## Male-biased protein dosage of Z-specific genes

We tested for putative post-transcriptional mechanisms by assessing the dosage compensation pattern at the proteomic level in adult *S. mansoni*, using a somatic tissue (head region) as well as the gonads; three replicates were used for each tissue and sex. Heads and gonads were chosen as they allowed us to compare Z-specific gene dosage in tissues with widespread functional sex-specificity (ovary and testis), and in a tissue where most dosage imbalances are likely to be deleterious. We used a label-free quantitative mass spectrometry approach to obtain a relative quantification of the protein levels in each tissue depending of the sex (*Figure 4—source data 1* to 5). Post-transcriptional dosage compensation mechanisms would be detectable by (i) an equalization of the Z expression between sexes at the protein level (F:M close to one for both the Z and the autosomes); (ii) a different correlation between F:M obtained from mRNA and from proteins for Z-linked and autosomal genes. We used publicly available head and gonad microarray data (*Nawaratna et al., 2011*) as the transcriptomic reference (*Figure 4—source data 6*). A significant and positive correlation was found between the F:M ratio derived from the microarray and from the proteomic data (*Figure 4*), and between transcript and protein dosage levels in both males and females (*Figure 4—figure supplements 1* and *2*), confirming the validity of the comparison.

Similar to what was observed using RNA-seq, the expression of Z-specific genes was strongly male-biased compared to that of autosomal genes in both heads (F:M of 0.68 for the Z chromosome versus 0.92 for the autosomes; *Figure 4A*, *Supplementary file 1*) and gonads (F:M of 0.78 versus 0.99; *Figure 4B*, *Supplementary file 1*). These F:M ratios are closer to each other than in our RNA-seq analysis (*Figure 2*, *Supplementary file 1*), or than the microarray data (*Supplementary file 1*), which could suggest a potential contribution of post-transcriptional regulation to dosage equalization. However, *Figure 4B* shows that Z-linked and autosomal genes show a similar correlation between the F:M ratios found for mRNAs and proteins (p>0.05 with a Fisher r-to-z transformation of the correlation coefficients, *Figure 4C and D*), which argues against a major role of post-transcriptional regulation to balance expression. This similarity between Z-linked and autosomal genes holds when only genes with male-biased expression in the microarray data are considered (*Figure 4—figure supplement 3*), and when the transcript and protein dosage of Z-linked autosomal genes are compared within each sex (*Figure 4—figure supplements 1* and *2*).

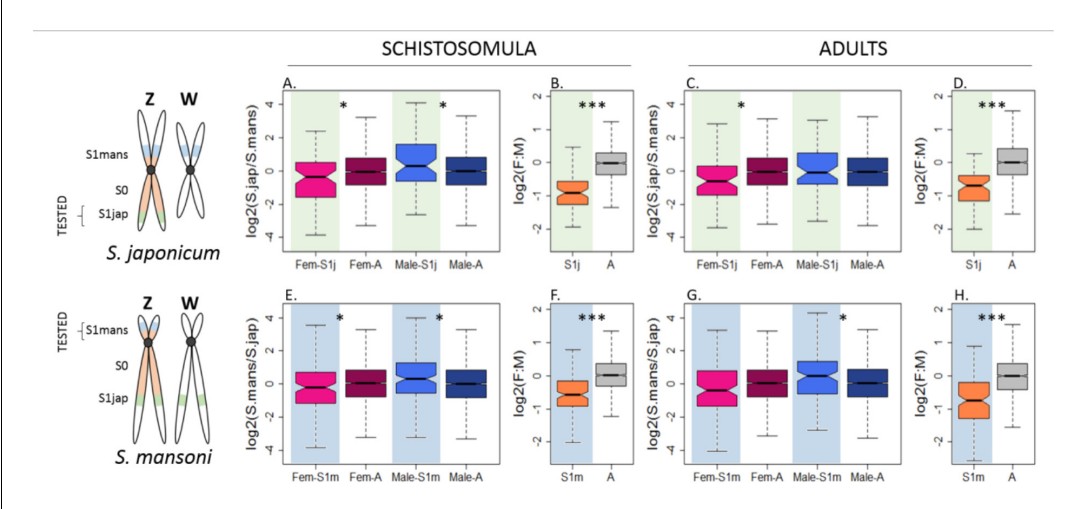

**Figure 3.** Convergent changes in the expression of Z-linked genes in *S. japonicum* and *S. mansoni* after sex chromosome differentiation. The Z-linked and autosomal gene expression patterns (normalized by ancestral pseudoautosomal or autosomal expression, to show changes since the appearance of the two S1 strata) are shown for S1jap in *S. japonicum* (A-D) and S1mans in *S. mansoni* (E-F). Fem-S1j and Male-S1j refer to the normalized expression levels of genes in the stratum S1jap in females and males, Fem-S1m and Male-S1m refer to the normalized expression levels in S1mans, and Fem-A and Male-A refer to the normalized expression levels of autosomal genes in females and in males, respectively. The level of significance of each comparison (Wilcoxon rank sum test with continuity correction) is denoted with asterisks: *p-value<0.05, **p-value<0.001, ***p-value<0.0001.
DOI: https://doi.org/10.7554/eLife.35684.029

The following source data and figure supplements are available for figure 3:

**Source data 1.** One-to-one orthology *S. mansoni* vs *S. japonicum.*
DOI: https://doi.org/10.7554/eLife.35684.032
**Source data 2.** Transcriptomic data, for blat 1-to-1 orthologs.
DOI: https://doi.org/10.7554/eLife.35684.033
**Source data 3.** Transcriptomic data, for WormBase Biomart orthologs.
DOI: https://doi.org/10.7554/eLife.35684.034
**Figure supplement 1.** Z-linked and autosomal female:male ratio of gene expression, normalized by ancestral autosomal expression, and using different filters.
DOI: https://doi.org/10.7554/eLife.35684.030
**Figure supplement 2.** Z-linked and autosomal gene expression in females and males, normalized by ancestral autosomal expression, and using different filters.
DOI: https://doi.org/10.7554/eLife.35684.031

## Discussion

### Schistosome sex chromosome evolution in the age of genomics

*S. mansoni*, *S. haematobium* and *S. japonicum* are the main species responsible for human schistosomiasis and have been the subject of many molecular and genomic studies. Despite the availability of extensive genomic and transcriptomic resources (e.g. a genome assembly at the near-chromosome level for *S. mansoni* (*Protasio et al., 2012a*; *Criscione et al., 2009*), or sex- and stage-specific transcriptomes (*Picard et al., 2016*; *Lu et al., 2016*; *Grevelding et al., 2018*; *Lu et al., 2017*), many basic questions remain regarding their reproduction and biology. For instance, the master sex-determining gene (and whether it is located on the W or Z) is still a mystery (*Lepesant et al., 2012*; *Portela et al., 2010*). This is partly due to the inherent challenges of assembling genomes from sequencing data, especially for regions rich in heterochromatin and repetitive sequences, such as sex chromosomes. For instance, 416 scaffolds, including 3893 genes (29% of the annotated nuclear genes), are still unplaced. By basing our analysis on genomic coverage, we were able to detect a further 285 Z-specific genes in *S. mansoni*; their role in sex determination can be investigated further. Our comparative approach can also reduce the number of candidates, as any gene involved in sex determination should in principle be found in the ancestral Z-specific stratum; similar analyses in

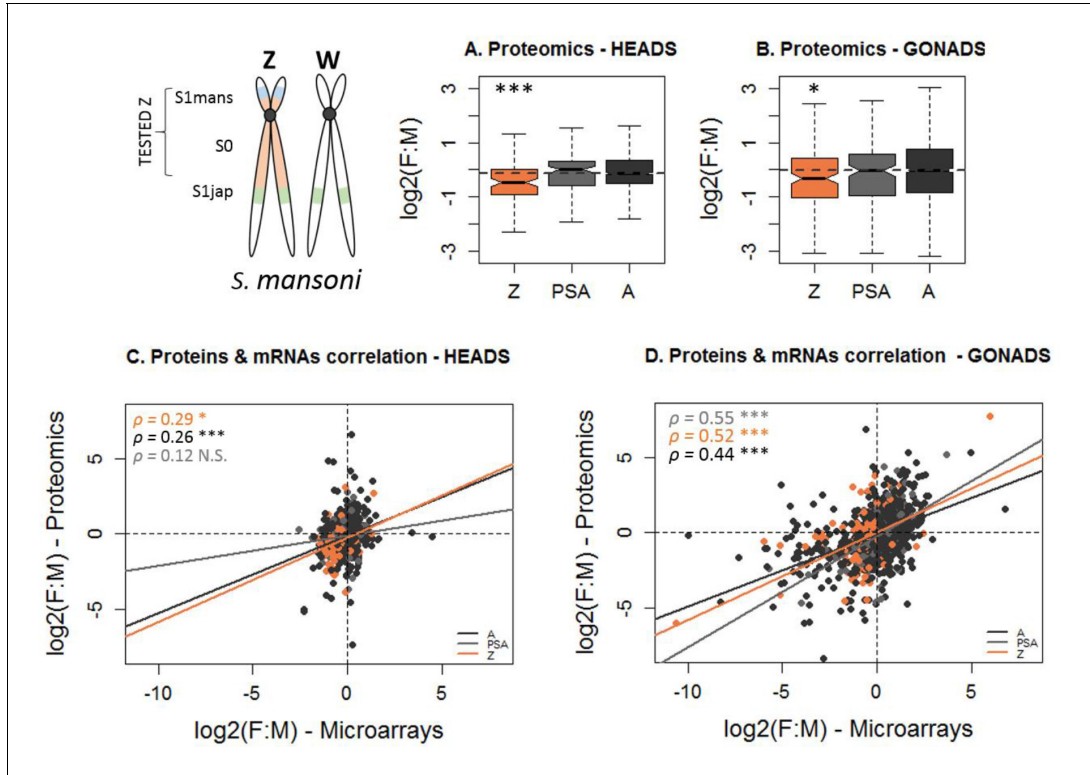

**Figure 4.** Transcript and protein dosage of Z-linked and autosomal genes in *S.mansoni* heads and gonads. (A), (B) The female:male (F:M) ratio of protein dosage in *S. mansoni* heads (A) and gonads (B), for Z-specific (Z), pseudoautosomal (PSA) or autosomal (A) genes. The dotted line shows the autosomal median of F:M expression. The level of significance of each comparison (Wilcoxon rank sum test with continuity correction) is denoted with asterisks: *p-value<0.05, **p-value<0.001, ***p-value<0.0001. (C), (D) Pearson correlation between the female:male ratio of expression obtained by proteomics (y-axes) and by microarrays (x-axes) in *S.mansoni* heads (C) and gonads (D). A positive correlation (coefficients $\rho$) is observed for Z-linked (Z, in orange), pseudoautosomal (PSA, in grey) and autosomal genes (A, in darkgrey). The level of significance of each correlation is denoted by asterisks: *p-value<0.05, **p-value<0.001, ***p-value<0.0001, N.S. p-value>0.05. No significant difference was found between the correlation obtained for Z-linked and autosomal genes in either tissue (using a Fisher r-to-z transformation).

DOI: https://doi.org/10.7554/eLife.35684.035

The following source data and figure supplements are available for figure 4:

**Source data 1.** Proteomic data with imputed values – GONADS.

DOI: https://doi.org/10.7554/eLife.35684.039

**Source data 2.** Proteomic data with imputed values – HEADS.

DOI: https://doi.org/10.7554/eLife.35684.040

**Source data 3.** Proteomic data without imputed values – GONADS.

DOI: https://doi.org/10.7554/eLife.35684.041

**Source data 4.** Proteomic data without imputed values – HEADS.

DOI: https://doi.org/10.7554/eLife.35684.042

**Source data 5.** Correspondance between Gene_Id and Protein_ID.

DOI: https://doi.org/10.7554/eLife.35684.043

**Source data 6.** Microarray data.

DOI: https://doi.org/10.7554/eLife.35684.044

**Figure supplement 1.** Pearson correlations between gene dosage at the transcript and protein levels in male heads and gonads.

DOI: https://doi.org/10.7554/eLife.35684.036

**Figure supplement 2.** Pearson correlations between gene dosage at the transcript and protein levels in female heads and gonads.

DOI: https://doi.org/10.7554/eLife.35684.037

**Figure supplement 3.** Pearson correlations between the female:male ratio of expression obtained by proteomics (y-axes) and by microarrays (x-axes) in *S. mansoni* heads and gonads, using only genes with male-biased expression in the microarray data.

DOI: https://doi.org/10.7554/eLife.35684.038

other species can in the future refine the candidate region. Another advantage of basing our Z-assignment purely on coverage patterns is that our results should be largely independent of potential biases in the current version of the genome. It should, however, be noted that many genes are likely still missing from the current assembly (which has a BUSCO score of 76% complete plus fragmented genes; https://parasite.wormbase.org/index.html [*Howe et al., 2016*; *Howe et al., 2017*]) and that repeating these analyses using future improved assemblies will be necessary to obtain the full set of sex-linked genes.

A gradient of ZW heteromorphism between schistosome species was revealed by cytogenetic studies (*Grossman et al., 1981a*; *Grossman et al., 1981b*; *Short and Grossman, 1981*); in particular, African schistosomes were found to have much more extensive ZW differentiation and W heterochromatinization than Asian species (*Hirai et al., 2000*). Our results generally support these cytogenetic data: we confirm the acquisition of independent evolutionary strata in the sex chromosomes of *S. mansoni* and *S. japonicum*, and detect a larger number of Z-specific genes in the African species (8% to 11% of all annotated orthologs, respectively, in *S. mansoni* and *S. haematobium*) than in *S. japonicum* (5.5% of all annotated orthologs). Interestingly, although the sex chromosomes of the African *S. mansoni* and *S. haematobium* differ morphologically, they are largely similar in their gene content (*Figure 1*), consistent with their much closer phylogenetic relationship (the median synonymous divergence between the two species is around 17%, compared to 65% for *S. mansoni/S. japonicum*, Appendix 1). This may be comparable to snakes, where ZW pairs with vastly different morphologies were all equally differentiated at the genomic level (*Vicoso et al., 2013a*), and highlights the contribution of other factors, such as differential transposable element accumulation, to the large-scale morphology of sex chromosomes.

## ZW systems and incomplete dosage compensation: gene-by-gene or partial shift?

ZW systems (aside from Lepidoptera) consistently show male-biased expression of the Z chromosome (*Gu and Walters, 2017*). While female-biased expression of the X occurs in a few young XY systems (*Gu and Walters, 2017*; *Schultheiß et al., 2015*; *Howe et al., 2017*; *Grossman et al., 1981b*; *Short and Grossman, 1981*; *Hirai et al., 2000*; *Mank and Ellegren, 2009*), well-established X chromosomes generally show full equalization of gene expression between the sexes. This difference has often been framed as the acquisition of global mechanisms of dosage compensation, which affects the whole X/Z, versus the acquisition of local compensation, in which dosage-sensitive genes become individually regulated (*Mank and Ellegren, 2009*). Several parameters should influence this, and favor local compensation in ZW systems: (i) The speed of the heterochromosome degeneration: when only a few genes are lost at a time (because the region of suppressed recombination is small, or because degeneration is slow), the establishment of a gene-by-gene dosage compensation may be favored; on the other hand, the loss of many genes at once could favor global mechanisms of dosage compensation (*Gu and Walters, 2017*; *Vicoso and Charlesworth, 2009b*). Since more mutations occur during spermatogenesis than oogenesis, female-specific W chromosomes will generally have lower mutation and degeneration rates than male-specific Ys, favoring local compensation; (ii) The effective population size of Z (NeZ): NeZ is decreased when the variance in reproductive success of ZZ males is larger than that of ZW females (e.g. in the presence of strong sexual selection). This will impair the adaptive potential of the Z (*Mank, 2009*; *Mullon et al., 2015*), such that only strongly dosage-sensitive genes can become upregulated in the heterogametic sex, while the others remain uncompensated; (iii) More efficient purging of mutations that are deleterious to males: strong sexual selection can also increase the strength of purifying selection on males, by preventing all but the fittest males from contributing to the next generations. If mutations that compensate for the loss of Y/W-linked genes overexpress the X/Z copy in both sexes, they will be under negative selection in the homogametic sex, and may be more efficiently selected against when males are the homogametic sex (*Mullon et al., 2015*).

Schistosomes are unusual among female-heterogametic clades in that they appear to have a chromosome-wide upregulation of the Z in both sexes; such an increase in males was not detected in birds (*Julien et al., 2012*) or snakes (*Vicoso et al., 2013a*), even when ancestral expression was taken into account. They therefore likely represent an intermediate between ZW species with true local compensation, and the chromosome-wide compensation of the ZW Lepidoptera. These results further show that, even if mutations that upregulate gene expression in both sexes are more easily

fixed on an evolving X-chromosome than on an evolving Z (*Mullon et al., 2015*), this is not an absolute barrier to the evolution of global dosage compensation. It is however still unclear why the evolutionary dynamics appear to differ between schistosomes and most other ZW clades, as the demographic and population genetics parameters of this group are largely unknown. The observed male biased sex-ratio in adults, combined with a largely monogamous mating system (*Beltran and Boissier, 2009*; *Beltran and Boissier, 2008*), may increase the reproductive variance of males and could reduce the effective population size of the Z. This should also lead to stronger sexual selection in males than in females (*Beltran and Boissier, 2008*; *Steinauer et al., 2009*), suggesting similar evolutionary dynamics as in other ZW systems. A detailed characterization of the population genetics of the Z chromosome and autosomes will therefore be crucial for understanding what may have driven the evolution of this unusual system.

## The relevance of the Ohno's hypothesis in the high-throughput sequencing era

Ohno's hypothesis predicts that the heterochromosome is initially overexpressed in both sexes, then secondarily downregulated in the homogametic sex (*Ohno, 1967*). This theoretical scenario was first formulated to account for the inactivation of the X in mammals. Since then, similar molecular mechanisms to downregulate the X/Z chromosome have been characterized in nematodes and moths (*Kiuchi et al., 2014*; *Meyer, 2010*). If an initial upregulation of the X did occur in both sexes, then inactivation in the homogametic sex should simply restore the ancestral expression levels, a hypothesis that has been tested in many empirical studies in mammals. Most of them assumed that the X and autosomes must have had similar ancestral levels of expression, and simply compared their expression (*Julien et al., 2012*; *Nguyen and Disteche, 2006*; *Xiong et al., 2010*; *Gu and Walters, 2017*; *Chen and Zhang, 2015*; *Graves, 2016*; *Deng et al., 2011*; *Kharchenko et al., 2011*; *Yildirim et al., 2011*; *Lin et al., 2012*; *Pessia et al., 2014*). These yielded mixed results, with some (*Xiong et al., 2010*) finding reduced expression of the X, while others (e.g. *Deng et al., 2011*; *Kharchenko et al., 2011*; *Yildirim et al., 2011*) found similar levels of expression for X-linked and autosomal genes, in agreement with Ohno's predictions. Taking ancestral gene expression into account, *Julien et al. (2012)* found evidence of an Ohno-like mechanism in the marsupials but not in placental mammals (*Julien et al., 2012*). *Pessia et al. (2012)* recently found that while individual dosage-sensitive genes do show evidence of upregulation, the majority does not. The evolution of X-inactivation may therefore have involved a complex scenario under which a few dosage-sensitive genes first became individually upregulated in both sexes (gene-by-gene compensation), followed by the establishment of a chromosome-wide mechanism to downregulate expression in females (global compensation) (*Sangrithi et al., 2017*; *Pessia et al., 2012*).

Our results, which consider ancestral expression and do not indicate a major influence of post-transcriptional regulation, suggest a scenario closer to Ohno's original hypothesis, with the male Z showing a consistent increase in expression. A similar pattern has been observed in *Tribolium castaneum* (Coleoptera, *Prince et al., 2010*), where the female X has been found to be over-expressed relative to the autosomes, and to the male X-chromosome. However, an RNA-seq analysis in the same species did not detect this (*Mahajan and Bachtrog, 2015*), so it is at this point unclear whether it truly represents an example of Ohno's model in action. The youngest evolutionary stratum of the young XY pair of threespine sticklebacks also shows overexpression in females (*Schultheiß et al., 2015*), even when ancestral expression is accounted for (*White et al., 2015*). However, the interpretation of these patterns is complicated by the fact that such an overexpression is also detected for the pseudoautosomal region, and that the oldest evolutionary stratum appears to lack dosage compensation altogether. Schistosomes may therefore not only represent an ideal system in which to investigate the evolution of dosage compensation in a ZW system, but also an unparalleled system for understanding the relevance of the model and predictions originally made by Ohno.

## Materials and methods

A detailed description of the computational analyses, as well as all the scripts that were used, are provided in Appendix 1.

## DNA sequencing of *S. japonicum* males and females

Male and female worms preserved in ethanol of *S. japonicum* were provided by Lu Dabing from Soochow University (Suzhou, China). DNA was extracted from 28 pooled males and 33 pooled females. The worms were lysed using the Tissue Lyser II kit (QIAGEN) and DNA was isolated using the DNeasy Blood and Tissue Kit (QIAGEN). DNA was then sheared with Covaris Focused-ultrasonicator. Library preparation and sequencing (HiSeq 2500 v4 Illumina, 125 bp paired-end reads) were performed at the Vienna Biocenter Next Generation sequencing facility (Austria). Reads have been deposited at the NCBI Short Reads Archive under accession number SRP135770.

## Publicly available DNA reads and genome assemblies

*S. mansoni* DNA libraries (100 bp paired-end reads) were downloaded from the NCBI Sequence Read Archive, under the accession numbers ERR562989 (~6000 male pooled cercariae) and ERR562990 (~6000 female pooled cercariae). Female *S. haematobium* DNA libraries (80 bp paired-end reads) were found under accession numbers ERR037800 and ERR036251. No male *S. haematobium* library was available. The reference genome assemblies of *S. mansoni* (PRJEA36577, [*Protasio et al., 2012b*]), *S. haematobium* (PRJNA78265, [*Young et al., 2012*]) and *S. japonicum* (PRJEA34885, [*Zhou et al., 2009*]) were obtained from the WormBase parasite database (https:// parasite.wormbase.org/index.html [*Howe et al., 2016*; *Howe et al., 2017*]).

## Orthology and assignment to the *S. mansoni* chromosomes

*S. mansoni* coding sequences and their respective chromosomal locations were obtained from the WormBase Parasite database (https://parasite.wormbase.org/index.html, [*Howe et al., 2016*; *Howe et al., 2017*]). This gene set was mapped to the *S. haematobium* and *S. japonicum* genome assemblies using Blat (*Kent, 2002*) with a translated query and dataset (-dnax option), and a minimum mapping score of 50; only the genome location with the best score was kept for each. When more than one gene overlapped by more than 20 base pairs, only the gene that had the highest mapping score was kept. Finally, each scaffold was assigned to one of the *S. mansoni* chromosomes, depending on the majority location of the genes that mapped to it, or on their total mapping scores if the same number of genes mapped to two separate chromosomes. The final chromosomal assignments are provided in *Figure 1—source data 1*.

## DNA read mapping and estimation of genomic coverage

For the *S. japonicum* DNA reads, adaptors were removed using Cutadapt (v1.9.1 [*Martin, 2011*]) and the quality of the reads was assessed using FastQC (v0.11.2, https://www.bioinformatics.babra-ham.ac.uk/projects/fastqc/); no further quality trimming was deemed necessary. For the *S. mansoni* and *S. haematobium* reads, potential adaptors were systematically removed, and reads were trimmed and filtered depending on their quality, using Trimmomatic (v0.36 [*Bolger et al., 2014*]). The resulting read libraries of each species were mapped separately to their reference genomes using Bowtie2 (–end-to-end –sensitive mode, v2.2.9 [*Langmead and Salzberg, 2012*]). The resulting alignments were filtered to keep only uniquely mapped reads, and the male and female coverages were estimated from the filtered SAM files with SOAPcoverage (v2.7.7., http://soap.genomics.org. cn/index.html). Coverage values were calculated for each scaffold in *S. haematobium* and *S. japonicum*, and for each 10 kb non-overlapping window in *S. mansoni*. The coverage values for each library are provided in *Figure 1—source data 3* and *4*.

## Detection of Z-specific sequences

For each species, we calculated the log2(female:male) coverage of each scaffold or, in the case of *S. mansoni*, of each 10 Kb window along the genome. Since only female DNA data was available for *S. haematobium*, the log2(female1 +female2) was used instead for this species.

In order to determine the 95% and 99% percentile of log2(female:male) of Z-linked sequences, which we use as cutoff values for assignment to Z-specific regions, we first excluded scaffolds/windows that fit an autosomal profile. To do so, the 1st and 5th percentile of log2(female:male) were estimated using all 10 kb windows found on the annotated autosomes of *S. mansoni* (*Protasio et al., 2012b*); in *S. haematobium* and *S. japonicum*, all scaffolds that mapped to the *S. mansoni* autosomes were used for this purpose.

By plotting the distribution of log2(female:male) for the scaffolds/windows that map to each chromosome (Appendix 1), we determined that: (i) for *S. mansoni* the 1st percentile (log2(female:male) =−0.26) discriminates effectively between autosomal and Z-specific windows; (ii) for *S. haematobium* and *S. japonicum*, which have noisier coverage, the 1st percentile included a significant fraction of Z-derived genes, and the 5st percentile was used instead (respectively log2(female) = 4.57 and log2 (female:male)=−0.40). All scaffolds with higher log2(female:male) or log2(female) were excluded.

The 95th and 99th quantiles of coverage were then calculated for the remaining, putatively Z-linked, sequences. By plotting all the coverage values along the *S. mansoni* Z-chromosome (Appendix 1 and *Figure 1*), we determined that: (i) for *S. mansoni* and *S. haematobium*, the 95th quantile (log2(female:male)=−0.4 and log2(female) = 4.41, respectively) was an effective cut-off for discriminating pseudoautosomal and Z-specific sequences; (ii) for *S. japonicum*, using the 95th percentile lead to the exclusion of many genes in Z-specific regions, and the 99st percentile (log2(F:M) =−0.84) was used instead. The Z-specific or autosome assignation was finally attributed as follows: (i) in *S. mansoni*, windows were classified as Z-linked if they displayed log2(F:M)<−0.4 and as autosomal if log2(F:M)>−0.4; (ii) for *S. haematobium*, scaffolds with log2(femalesum) <4.41 were classified as Z-linked, scaffolds with log2(femalesum) >4.57 as autosomal, and all others as ambiguous; (iii) for *S. japonicum*, scaffolds with log2(F:M)<−0.84 were classified as Z-linked, scaffolds with log2(F:M) >−0.40 as autosomal, and all others as ambiguous. For *S. haematobium* and *S. japonicum*, we considered only scaffolds with at least a coverage of 1 in each library (n.a.). In *S. mansoni*, unplaced scaffolds shorter than 50 kb were excluded; five consecutive 10 kb windows with consistent coverage patterns were required for a region to be classified as either Z-specific or autosomal. Smaller regions, as well as the two 10 Kb windows surrounding them, were excluded (see Appendix 1). The final classifications for the three species are provided in *Figure 1—source data 2*, *3* and *4*, and summarized for orthologs in *Figure 1—source data 1*.

## Definition of Z-specific strata in *S. mansoni* and *S. japonicum*

The Z-specific gene content of *S. mansoni* and *S. japonicum* was compared in order to define Z-chromosome strata. Genes that were located on Z-specific scaffolds/windows in both species were assigned to the shared stratum 'S0'. Genes that were assigned to Z-specific regions in one species but not in the other were assigned to lineage-specific strata: 'S1mans' genes were Z-specific in *S. mansoni* and autosomal in *S. japonicum*, while 'S1jap' genes were Z-specific in *S. japonicum* and autosomal in *S. mansoni*. While the main figures consider all the genes that were classified as Z-linked or autosomal based on coverage (referred to as the 'exhaustive classification' in Appendix 1 and Figures), independent of their original genomic location, we repeated the analyses using only the Z-specific genes that were already assigned to the ZW linkage map of *S. mansoni* (the 'stringent classification' in Appendix 1). All genes belonging to the categories 'excluded', 'ambiguous', 'n.a.' or that did not have orthologs on *S. japonicum* scaffolds were not further considered (*Table 1*). 'PSA_shared' and 'Aut_shared' are common to the two classifications and correspond to genes that were classified as autosomal in both species using coverage and that were previously mapped to the ZW linkage group or to the autosomes of *S. mansoni*, respectively (*Protasio et al., 2012b*).

## Publicly available RNA reads and estimation of gene expression

*S. mansoni* and *S. japonicum* RNA-seq libraries were obtained from SRA (NCBI). Accession numbers are: *S. mansoni* adult females: ERR506076, ERR506083, ERR506084; *S. mansoni* adult males: ERR506088, ERR506082, ERR506090; *S. mansoni* schistosomula females: SRR3223443, SRR3223444; *S. mansoni* schistosomula males: SRR3223428, SRR3223429; *S. japonicum* adult females: SRR4296944, SRR4296942, SRR4296940; *S. japonicum* adult males: SRR4296945, SRR4296943, SRR4296941; *S. japonicum* schistosomula females: SRR4279833, SRR4279491, SRR4267990; *S. japonicum* schistosomula males: SRR4279840, SRR4279496, SRR4267991. Raw reads were cleaned using trimmomatic (v 0.36 [*Bolger et al., 2014*]), and the quality of the resulting reads was assessed using FastQC (v0.11.2, https://www.bioinformatics.babraham.ac.uk/projects/fastqc/). Reads were mapped to their respective reference genomes used Tophat2 (*Trapnell et al., 2009*). Read counts were obtained with H (*Anders et al., 2015*) and expression values (in Reads Per Kilobase of transcript per Millionmapped reads, RPKM) were calculated for each gene in each of the RNA-seq libraries (*Figure 2—source data 1*). TPM (Transcripts Per Kilobase Million) values were also calculated using

Kallisto (*Bray et al., 2016*) against set of coding sequences of the respective species. All expression values are provided in *Figure 3—source data 2* and *3*. A Loess Normalization (R library Affy) was performed on the schistosomulum data and on the adult data separately and all analyses were performed using different thresholds (RPKM > 0, RPKM > 1, TPM >0 or TPM >1 in all the libraries of the studied stage). The Loess normalization was applied to all conditions at once when we filtered for minimum expression in all stages and sexes (RPKM > 1 and RPKM > 3, *Figure 2—figure supplements 15–17*, *Figure 3—figure supplements 1* and *2*). Correlation analyses were performed for each developmental stage, considering libraries from both males and females, and both species. As shown in Appendix 1, two *S. mansoni* libraries (ERR506076 and ERR506082) were not well correlated with the other samples, and were excluded from our study. Expression values were averaged for each stage and sex. The significance of differences between medians of expression was tested with Wilcoxon rank sum tests with continuity correction.

## Detection of *S. mansoni* and *S. japonicum* one-to-one orthologs

*S. japonicum* coding DNA sequences and their respective location on the genome scaffolds were obtained from the WormBase Parasite database (https://parasite.wormbase.org/index.html [*Howe et al., 2016*; *Howe et al., 2017*]). The *S. mansoni* set of coding sequences (see above) was mapped to the *S. japonicum* gene set using Blat (*Kent, 2002*) with a translated query and dataset (-dnax option), and a minimum mapping score of 50; only reciprocal best hits were kept. This reciprocal best hit ortholog list is provided in *Figure 3—source data 1* and *2*. A second list of orthologs was obtained from the Biomart of WormBase Parasite, excluding paralogues, and requiring a gene stable ID for both *S. mansoni* (PRJEA36577) and *S. japonicum* (PRJEA34885) (in *Figure 3—source data 1* and *3*). In subsequent transcriptomic analyses, each list was used independently to ensure that the results were independent of the method used to assign orthology.

## Microarray analysis

Microarray data for male and female heads and gonads (*Nawaratna et al., 2011*) were obtained from the Gene Expression Omnibus (GEO) database (NCBI, ftp://ftp.ncbi.nlm.nih.gov/geo/series/GSE23nnn/GSE23942/matrix/). A Loess Normalization (R library Affy) was performed on the head and gonad data separately. When different probes corresponded to one gene, their expression values were averaged. Gene expression was available for a total of 6925 genes. The normalized data are available in *Figure 4—source data 6*.

## Protein extraction and proteomic analysis

Male and female adult *S. mansoni* gonads were sampled using the whole-organ isolation approach described previously (*Hahnel et al., 2013*). Twenty ovaries and 20 testes, as well as five heads of each sex, were sampled, in triplicate, from paired worms. All biological samples were resuspended in Laemmli buffer, denatured and frozen at −20°C until further processing. Subsequent protein treatment and analyses were performed at the 'EDyP-service' – proteomic platform (Grenoble, France). The extracted proteins were digested by modified trypsin (Promega, sequencing grade). The resulting peptides were analyzed by nanoLC-MS/MS (Ultimate 3000 RSLCnano system coupled to Q-Exactive Plus, both Thermo Fisher Scientific). Separation was performed on a 75 µm x 250 mm C18 column (ReproSil-Pur 120 C18-AQ 1.9 µm, Dr. Maisch GmbH) after a pre-concentration and desalting step on a 300 µm × 5 mm C18 precolumn (Pepmap, Thermo Fisher Scientific).

MS and MS$^2$ data were acquired using Xcalibur (Thermo Fisher Scientific). Full-scan (MS) spectra were obtained from 400 to 1600 m/z at a 70,000 resolution (200 m/z). For each full-scan, the most intense ions (top 10) were fragmented in MS$^2$ using high-energy collisional dissociation (HCD). The obtained data were processed in MaxQuant 1.5.8.3 against the database loaded from Uniprot (taxonomy *Schistosoma mansoni*, October 26th, 2017, 13.521 entries) and the MaxQuant embedded database of frequently observed contaminants. The resulted iBAQ values (*Tyanova et al., 2016*) were loaded into ProStaR (*Wieczorek et al., 2017*) for statistical analysis. Contaminant and reverse proteins were removed and only the proteins with three quantified values in at least one condition were taken into account.

After log2 transformation, the iBAQ values were normalized by overall-wise median centering followed by imputation using detQuantile algorithm with quantile set to 1 (*Figure 4—source data 1*

and *2*). An alternative set of data without imputed values is available in *Figure 4—source data 3* and *4*. 1988 and 2750 *Schistosoma mansoni* proteins were identified in heads and in gonads, respectively (see *Figure 4—source data 1* and *2* for statistical testing of differential abundance between male and female samples). Among them, 1741 and 2516 could be attributed unambiguously to a *Schistosoma mansoni* gene and were represented by more than one peptide; these were subsequently analyzed (See *Figure 4—source data 5*).

## Data availability

DNA reads of male and female *S. japonicum* are available on the SRA database under study number SRP135770. Sex and tissue-specific *S. mansoni* label-free proteomic data are provided in *Figure 4—source data 1* to *Figure 4—source data 4*.

## Code availability

The full bioinformatic pipeline used in this study is provided in Appendix 1.

# Acknowledgements

We are grateful to Lu Dabing (Soochow University, Suzhou, China) for providing *Schistosoma japonicum* samples, to Ariana Macon (IST Austria) and Georgette Stovall (JLU Giessen) for technical assistance, to IT support at IST Austria for providing optimal environment to bioinformatic analyses, and to the Vicoso lab for comments on the manuscript. We thank the support of the discovery platform and informatics group at EDyP. Proteomic experiments were partly supported by the Proteomics French Infrastructure (ANR-10-INBS-08–01 grant) and Labex GRAL (ANR-10-LABX-49–01); and the Wellcome Trust, grant 107475/Z/15/Z to CGG and TQ (FUGI). This project was funded by an Austrian Science Foundation FWF grant (Project P28842) to BV.

# Additional information

## Funding

| Funder | Grant reference number | Author |
| --- | --- | --- |
| Austrian Science Fund | P28842 | Beatriz Vicoso |
| Proteomics French Infrastructure | ANR-10-INBS-08-01 | Yohann Couté |
| Labex GRAL | ANR-10-LABX-49-01 | Yohann Couté |
| Wellcome Trust | 107475/Z/15/Z | Thomas Quack Christoph Grevelding |

The funders had no role in study design, data collection and interpretation, or the decision to submit the work for publication.

## Author contributions

Marion A L Picard, Conceptualization, Formal analysis, Investigation, Methodology, Writing—original draft, Writing—review and editing; Celine Cosseau, Conceptualization, Writing—review and editing; Sabrina Ferré, Christoph G Grevelding, Yohann Couté, Formal analysis, Methodology, Writing—review and editing; Thomas Quack, Methodology, Writing—review and editing; Beatriz Vicoso, Conceptualization, Supervision, Funding acquisition, Writing—original draft, Writing—review and editing

## Author ORCIDs

Marion A L Picard (ID) http://orcid.org/0000-0002-8101-2518
Yohann Couté (ID) http://orcid.org/0000-0003-3896-6196
Beatriz Vicoso (ID) http://orcid.org/0000-0002-4579-8306

## Decision letter and Author response

Decision letter https://doi.org/10.7554/eLife.35684.065

Author response https://doi.org/10.7554/eLife.35684.066

## Additional files

### Supplementary files

• Supplementary file 1. Comparison of the ratio of expression between the Z-linked and autosomal genes (Z:A), and comparison of expression between males and females (F:M) in the different species, stages and methods (Supplementary Table 1). Comparison of female:male ratios of expression (F:M) using microarray and proteomics (Supplementary Table 2)
DOI: https://doi.org/10.7554/eLife.35684.045

• Transparent reporting form
DOI: https://doi.org/10.7554/eLife.35684.046

### Data availability

Sequencing data have been deposited to the NCBI short reads archive (PRJNA432803). Proteomic dosage values as well as final versions of the processed datasets (genomic coverage, expression values, chromosomal assignments) have been deposited into the IST Austria Data Repository (http://dx.doi.org/10.15479/AT:ISTA:109).

The following datasets were generated:

| Author(s) | Year | Dataset title | Dataset URL | Database, license, and accessibility information |
|---|---|---|---|---|
| Vicoso B | 2018 | DNA reads of male and female S. japonicum | https://www.ncbi.nlm.nih.gov/bioproject/PRJNA432803/ | Publicly available at NCBI BioProject (accession no: PRJNA432803) |
| Vicoso B | 2018 | Proteomic dosage values as well as final versions of the processed datasets (genomic coverage, expression values, chromosomal assignments) | http://dx.doi.org/10.15479/AT:ISTA:109 | Publicly available via the IST data repository |

The following previously published dataset was used:

| Author(s) | Year | Dataset title | Dataset URL | Database, license, and accessibility information |
|---|---|---|---|---|
| Nawaratna SSK, McManus DP, Moertel L, Gobert GN, Jones MK | 2010 | Gene atlasing of digestive and reproductive tissues in Schistosoma mansoni | https://www.ncbi.nlm.nih.gov/geo/query/acc.cgi?acc=GSE23942 | Publicly available at the NCBI Gene Expression Omnibus (accession no: GSE23942) |

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

## Appendix 1

DOI: https://doi.org/10.7554/eLife.35684.047

# Computational analyses

## Supplementary codes and inputs

All input files, as well as additional Perl and R scripts are available at the the IST Austria data repository (http://dx.doi.org/10.15479/AT:ISTA:109).

## Publicly available reference genomes and annotations

All data were obtained at the WormBase Parasite database (https://parasite.wormbase.org/index.html; *Howe et al., 2016*; *Howe et al., 2017*).

*S.mansoni*
Genomic sequences: schistosoma_mansoni.PRJEA36577.WBPS9.genomic.fa
Annotation: schistosoma_mansoni.PRJEA36577.WBPS9.canonical_geneset.gtf
CDS: schistosoma_mansoni.PRJEA36577.WBPS9.CDS_transcripts.fa
*S.japonicum*
Genomic sequences: schistosoma_japonicum.PRJEA34885.WBPS9.genomic.fa
Annotation: schistosoma_japonicum.PRJEA34885.WBPS9.canonical_geneset.gtf
CDS: schistosoma_japonicum.PRJEA34885.WBPS9.CDS_transcripts.fa
*S.haematobium*
Genomic sequences: schistosoma_haematobium.PRJNA78265.WBPS9.genomic.fa
Annotation: schistosoma_haematobium.PRJNA78265.WBPS9.canonical_geneset.gtf

## Orthology

The same pipeline was applied to *S. japonicum* and *S. haematobium*. Example is shown here for the *S. japonicum* species.

## Mapping of *S. mansoni* genes on the *S. japonicum* genome

First, *S. mansoni* gene sequences were mapped to the *S. japonicum* genome using Blat (*Kent, 2002*) with a translated query and database:

```
CDS     =     pathwaytosmansonigenome/schistosoma_mansoni.PRJEA36577.WBPS9.
CDS_transcripts.fa
GENOME =~ pathwaytojaponicumgenome
BLATOUTPUT =~ pathwaytoblatoutputfile
```

```
~/tools/blat -q = dnax -t = dnax -minScore = 50
${GENOME}/schistosoma_japonicum.PRJEA34885.WBPS9.genomic.fa ${CDS}
${BLATOUTPUT}/jap_mans.blat
```

The resulting blat alignment was filtered to keep only the mapping hit with the highest mapping for each *S. mansoni* gene:

```
sort --k 10 ${BLATOUTPUT}/jap_mans.blat > ${BLATOUTPUT}/jap_mans.blat.sorted
perl Script1_besthitblat.pl ${BLATOUTPUT}/jap_mans.blat.sorted
```

This produced the first filtered Blat alignment:

```
${BLATOUTPUT}/jap_mans.blat.sorted.besthit
```

In a second filtering step, when several *S. mansoni* genes overlapped on the *S. japonicum* genome by more than 20bps, we keep only the gene with the highest mapping score:

```
sort --k 14 ${BLATOUTPUT}/jap_mans.blat.sorted.besthit >
${BLATOUTPUT}/jap_mans.blat.sorted.besthit.sorted
perl  Script2_blatreverse.pl  ${BLATOUTPUT}/jap_mans.blat.sorted.besthit.
sorted
```

This produced the final filtered Blat alignment:
```
${BLATOUTPUT}/jap_mans.blat.sorted.besthit.sorted.nonredundant
```

## Assignment of syntenic blocks to *S. mansoni* chromosomal locations

Each *S. japonicum* scaffold was assigned to one of the *S. mansoni* chromosomes based on its gene content, using a majority rule.

(i) As input table, we transformed the output obtained in the previous step into a 3-column file (sorted by column 1): column1 = Smansoni_gene_ID, column2 = Sjaponicum_contig/scaffold, column3 = score.

```
cat ${BLATOUTPUT}/jap_mans.blat.sorted.besthit.sorted.nonredundant | cut -f
1,10,14 |
awk '{print $2, $3, $1}' | perl -pi -e 's/-P.//gi' | sort >
${BLATOUTPUT}/jap_mans.blat.sorted.besthit.sorted.nonredundant_score.
joignabl
```

(ii) A second file containing the chromosomal location of *S. mansoni* genes was also needed for the next step. The file (named here RefChromLocation.joignabl) is a 2-column file resulting from the combination of the *S. mansoni* GFF file, and the newly identified Z-specific genes (*i.e.* the chromosomal location corresponds to the published GTF location, except for 1. the new Z *versus* pseudoautosomal assignments based on coverage for ZW genes and 2. genes which were newly detected as Z-linked): column1 = Smansoni_gene_ID, column2 = Smansoni_chromosome. This file should be sorted by column 1. Column2 has 11 possible values: 'Chr_1', 'Chr_2', 'Chr_3','Chr_4', 'Chr_5', 'Chr_6', 'Chr_7', 'Chr_PSA', 'Chr_Z', 'Chr_newZ', and 'Chr_UP' (See *Figure 1—source data 1*, column #6: 'Bestloc_input_mans').

(iii) We joined both files by column1 and sort the joined table by Sjaponicum_contig/scaffold:

```
join ${BLATOUTPUT}/jap_mans.blat.sorted.besthit_score.joignabl pathwaytos-
mansonigenome/
RefChromLocation.joignabl | awk '{print $2, $1, $4, $3}' | sort >
jap_mans.inputforbestlocation
```

A final script counted how many orthologs from the different *S. mansoni* chromosomes were found in each *S. japonicum* scaffold, and assigned it a chromosome based on the location of the majority of the orthologs. If the same number of genes were located on two different chromosomes, then the chromosome for which the sum of the mapping scores was highest waschosen.

```
perl Script3_bestlocation.pl jap_mans.inputforbestlocation
```

## Identification of 1-to-1 orthologs between *S. mansoni* and *S. japonicum*

*S. japonicum* genes were first mapped to *S. mansoni* using Blat with a translated database and query:

```
CDS1=pathwaytosmansonigenome/schistosoma_mansoni.PRJEA36577.WBPS9.
CDS_transcripts.fa
```

```
CDS2=pathwaytojaponicumgenome/schistosoma_japonicum.PRJEA34885.WBPS9.
CDS_transcripts.fa
BLATOUTPUT =~ pathwaytoblatoutputfile
```

```
~/tools/blat -q = dnax -t = dnax -minScore = 50 ${CDS2} ${CDS1}
${BLATOUTPUT}/jap_mans_onetoone.blat
```

Only best reciprocal hits were kept:
```
perl Script4_Blat_one2one.pl ${BLATOUTPUT}/jap_mans_onetoone.blat
```

The list of orthologs is provided in *Figure 3—source data 1*.

## Determination of Z-specific regions in each species

### List of DNA libraries used

Species | Accession number | Library name
S. haematobium | ERR037800 | Female1
S. haematobium | ERR036251 | Female2
S. japonicum | 40640 | Male
S. japonicum | 40641 | Female
S. mansoni | ERR562989 | Male
S. mansoni | ERR562990 | Female

### Raw read processing

The same pipeline was applied to all the species. Example is shown here for the *S. japonicum* species.

The newly sequenced *S. japonicum* reads were provided as BAM files, which we first converted to FASTQ:

```
READS =~ pathwaytoreads
```

```
for i in 'ls ${READS} | grep. bam'; do
echo ${i}
~/tools/bedtools0.2.25.0/bin/bamToFastq -i ${i} -fq ${i}.fastq1 -fq2 ${i}.
fastq2
done
```

Adapter sequences were removed from the newly sequenced libraries (Only for *S. japonicum*) using Cutadapt (*Martin, 2011*):

```
LIBRARY_1 = Forward reads in fastq format
LIBRARY_2 = Reverse reads in fastq format
READS =~ pathwaytoreads
CUTADAPTOUTPUT =~ pathwaytocutadaptoutput
```

```
~/tools/cutadapt_1.9.1 --match-read-wildcards -f fastq -a
 AGATCGGAAGAGCACACGTCTGAACTCCAGTCAC ${READS}/${LIBRARY_1} -o
${CUTADAPTOUTPUT}/${LIBRARY_1}.trimmed
```

```
~/tools/cutadapt_1.9.1 --match-read-wildcards -f fastq -a
 AGATCGGAAGAGCGTCGTGTAGGGAAAGAGTGTA ${READS}/${LIBRARY_2} -o
${CUTADAPTOUTPUT}/${LIBRARY_2}.trimmed
```

Trimmomatic was used to remove adaptor and low quality sequences from the available libraries (For *S. mansoni* and *S. haematobium*, accession numbers are given in the main text):

```
READS =~ pathwaytoreads
LIBRARY = LibraryName

module load java
 java -jar ~/tools/tools/Trimmomatic-0.36/trimmomatic-0.36.jar PE -phred33
 ${READS}/${LIBRARY}.fastq1 ${READS}/${LIBRARY}.fastq2
   ${READS}/${LIBRARY}.fastq1_paired.fq.gz  ${READS}/${LIBRARY}.fastq1_un-
paired.fq.gz
${READS}/${LIBRARY}.fastq2_paired.fq.gz    ${READS}/${LIBRARY}.fastq2_un-
paired.fq.gz
ILLUMINACLIP:~/tools/Trimmomatic-0.36/adapters/TruSeq2-PE.fa:2:30:10
HEADCROP:12
 LEADING:3 TRAILING:3 SLIDINGWINDOW:4:15 MINLEN:36
```

All libraries were checked using FastQC:

```
for i in 'ls ${CUTADAPTOUTPUT} | grep. trimmed'; do
~/tools/FastQC-v0.11.2/fastqc ${CUTADAPTOUTPUT}/${i} --extract --o ~/fastQ-
Coutputs/
done
```

OR

```
for i in 'ls ${READS} | grep paired.fq.gz'; do
~/tools/FastQC-v0.11.2/fastqc ${READS}/${i} --extract --o ~/fastQCoutputs/
done
```

## Read alignment on reference genome
The same pipeline was applied to all the species. Example is shown here for the *S. japonicum* species.

Reads were mapped to the respective reference genomes using Bowtie2 (**Langmead and Salzberg, 2012**).

```
#Built reference genome

GENOME =~ pathwaytojaponicumgenome

~/tools/bowtie2-2.2.9/bowtie2-build
${GENOME}/schistosoma_japonicum.PRJEA34885.WBPS9.genomic.fa ref_genome

# Mapping

SPE = Sjaponicum
LIBRARY = LibraryName
BOWTIE2OUTPUT =~ pathwaytobowtie2output
THREADS = 5

~/tools/bowtie2-2.2.9/bowtie2 -x ${GENOME}/refgenome -1
${CUTADAPTOUTPUT}/${LIBRARY_1}.trimmed -2 ${CUTADAPTOUTPUT}/${LIBRARY_2}.
trimmed --end-to-end --
sensitive -p ${THREADS} -S ${BOWTIE2OUTPUT}/${SPE}_${LIBRARY}.sam
```

## Genomic coverage

The same pipeline was applied to all the species. Example is shown here for the *S. japonicum* species.

Genomic coverage in the male and female samples was calculated using SOAPcoverage (v2.7.7., http://soap.genomics.org.cn/index.html), after filtering for uniquely mapped reads.

```
#Select only uniquely mapped reads

cd ${BOWTIE2OUTPUT}

for i in 'ls | grep. sam';do
echo ${i}
grep -vw XS:i ${i} > ${i}_unique.sam
done

#Obtain average genomic coverage for each scaffold

GENOME =~ pathwaytojaponicumgenome
SOAPCOVOUTPUT = pathwaytosoapcoverageoutput

 cd ${BOWTIE2OUTPUT}

for j in 'ls | grep _unique.sam'; do
echo ${j}
~/tools/soapcoverage.2.7.7/soap.coverage -sam -cvg -i ${j} -onlyuniq -refsin-
gle
 ${GENOME} -o ${SOAPCOVOUTPUT}/${j}.soapcov
done
```

For *S. mansoni*, the genomic coverage was not calculated per scaffold but for 10 kb windows:

```
GENOME =~ pathwaytomansonigenome
LIBRARY = LibraryName
SOAPCOVOUTPUT = pathwaytosoapcoverageoutput
WIN = 10000

cd ${BOWTIE2OUTPUT}

for j in 'ls | grep _unique.sam'; do
echo ${j}
~/tools/soapcoverage.2.7.7/soap.coverage -sam -cvg -i ${j} -onlyuniq -refsin-
gle
 ${GENOME} -window ${WIN} -o ${SOAPCOVOUTPUT}/${j}.soapcov
done
```

The final coverage values are part of the *Figure 1—source data 2* (*S. mansoni*), *Figure 1—source data 3* (*S. haematobium*) and *Figure 1—source data 4* (*S. japonicum*)

## Determination of Z-specific maximum coverage threshold for *S. mansoni* reference species

The maximum value of log2(female:male) coverage for Z-specific assignment in *S. mansoni* was determined using the R script Script5_Genomics_mansoni (Part I) with the input file InputFile1_Smansoni_SoapCov_By10kb_ForR, which contains the SoapCov outputs for male (ERR562989_Depth) and female (ERR562990_Depth).

This involved the following steps (shown in ***Appendix 1-figure 1***):

1. When plotted, the log2(female:male) ratio of coverage shows a bimodal distribution. The highest peak is distributed around 0 and corresponds to the windows that are localized on the known autosomes (Chromosomes 1 to 7) of the current version of the genome assembly (***Protasio et al., 2012a***) (PRJEA36577).
2. If we consider only the unimodal distribution of autosomal window coverage, the 5th percentile has a log2(female:male)=−0.26.
3. We use this value to exclude the pseudoautosomal windows (plotted in grey) of the ZW linkage group and consider only the non-pseudoautosomal windows (plotted in orange) for the next step.
4. The non-pseudoautosomal window (InputFile2_Z_10 kb_5pr100) coverage displays an unimodal distribution, and the 99th percentile as a value log2(female:male)=−0.4.
5. When visualizing log2(female:male) along the ZW linkage group, it appears that the log2(female:male)=−0.4 threshold allows the discrimination of the three known Z-specific regions (in orange) and pseudoautosomal regions (in grey). To finely define the Z-specific content, we systematically excluded windows for which the coverage value is not consistent with the adjacent windows: 5 consecutive 10 kb windows with consistent coverage were required to be considered either pseudoautosomal or Z-specific. For instance, yellow bands on the graph highlight regions of more than one window but less than five. They are tagged as 'Excluded' in ***Figure 1—source data 2***, as are the two adjacent windows.

We then applied this threshold value to all the genome, considering only scaffolds longer than 50 kb for classification. The final classification of each scaffold (as 'Z' or 'Autosome') is presented in ***Figure 1—source data 2***. Used R script is: Script1_Genomics_mansoni (Part II); and corresponding input file is: InputFile1_Smansoni_SoapCov_By10kb_ForR.

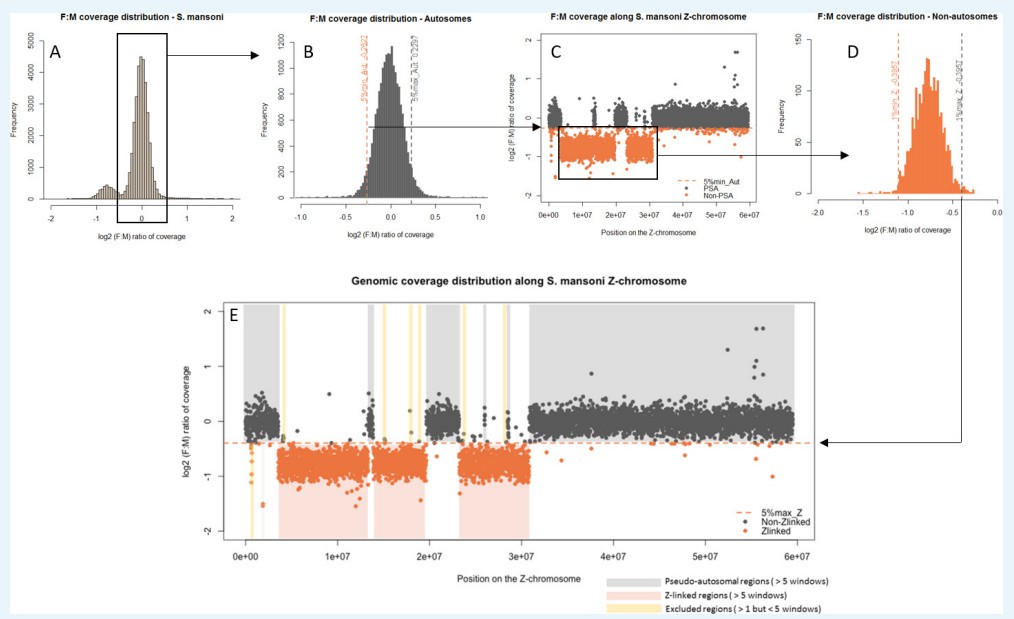

**Appendix 1—figure 1.** Determination of Z-specific maximum coverage threshold for S. mansoni reference species.
DOI: https://doi.org/10.7554/eLife.35684.048

All genes of a window were assigned to 'Z' or 'Autosome' depending on the window assignment. When a gene overlapped with an excluded window, it was excluded as well. In order to exclude these genes, the following script was used: Script6_GeneSelector, using as input a 3-column file (Chromosome ID, First base of window to exclude, Last base of window to exclude).

## Determination of Z-specific maximum coverage threshold for *S. japonicum*.

The maximum value of log2(female:male) coverage for Z-specific assignment in *S. japonicum* was determined using the R script Script7_Genomics_japonicum (Part I) with the input file InputFile3_Sjaponicum_SoapCov_Bestloc_ForR, which contains the SoapCov outputs for male (Depth_40640) and female (ERR562990_Depth), and the assignment to *S. mansoni* chromosomes (See paragraph 1.1).

This involved the following steps (shown in ***Appendix 1—figure 2***):

1. When plotted, the log2(female:male) ratio of coverage shows a bimodal distribution.
2. As *S. japonicum* genome is only assembled at the scaffold level (***Zhou et al., 2009***), we used the log2(female:male) coverage of the scaffold mapping to *S. mansoni* autosomes to get the distribution of autosomal log2(female:male).
3. If we consider only the unimodal distribution of the autosome-assigned scaffold coverage, the 1$^{st}$ percentile has a log2(female:male)=$-0.4$.
4. We use this value to classify scaffolds that mapped to the *S. mansoni* ZW linkage group as Z-specific or pseudoautosomal (grey in ***Appendix 1—figure 2***, panel D), and consider only the non-pseudoautosomal scaffolds (plotted in orange) for the next step.
5. The non-pseudoautosomal scaffold (InputFile4_Sjaponicum_1pr100) coverage displays an unimodal distribution, and the 95$^{th}$ percentile has a value of log2(female:male)=$-0.84$.
6. When visualizing the log2(female:male) along the ZW linkage group, it appears that no single threshold value can finely discriminate Z-specific and pseudoautosomal scaffolds. So we use the log2(female:male)=$-0.84$ threshold as maximum value for Z-specific regions (in orange), and log2(female:male)=$-0.4$ threshold as minimum value for pseudoautosomal regions (in grey). Scaffolds with log2(female:male) between these two values were classified as 'ambiguous'.

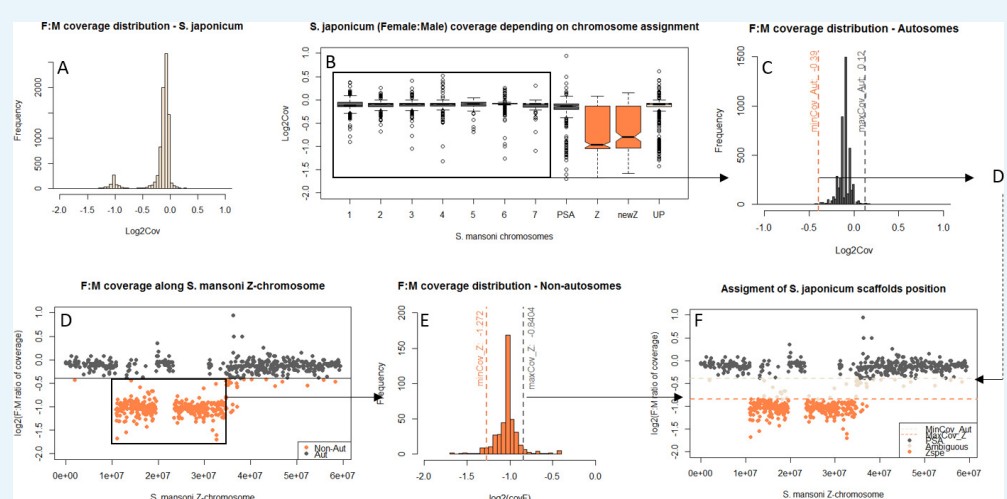

**Appendix 1—figure 2.** Determination of Z-specific maximum coverage threshold for S. japonicum species.
DOI: https://doi.org/10.7554/eLife.35684.049

We then applied this threshold value to all the genome. Used R script is: Script7_Genomics_japonicum (Part II); and corresponding input file is: InputFile3-2_Sjaponicum_SoapCov_ForR.

The final 'Z', 'Autosome' or 'Ambiguous' assignment is shown in *Figure 1—source data 4*.

## Determination of Z-specific maximum coverage threshold for *S. Haematobium*.

The maximum value of log2(Female) coverage for Z-specific assignment in *S. haematobium* was determined using the R script Script8_Genomics_haematobium (Part I) with the input file InputFile5_Shaematobium_SoapCov_Bestloc_ForR, which contains the SoapCov outputs for two female libraries (Depth_ERR037800, Depth_ERR036251), and the assignment to *S. mansoni* chromosomes (See paragraph 1.1).

This involved the following steps (shown in *Appendix 1—figure 3*):

1. When plotted, the log2(Female) coverage shows a bimodal distribution.
2. As the *S. haematobium* genome is only assembled at the scaffold level (*Young et al., 2012*), we used the log2(Female) coverage of the scaffold mapping to *S. mansoni* autosomes to get the distribution of autosomal log2(Female).
3. If we consider only the unimodal distribution of the autosome-assigned scaffold coverage, the 5th percentile has a log2(Female) = 4.4
4. We use this value to classify scaffolds that mapped to the *S. mansoni* ZW linkage group as Z-specific or pseudoautosomal (grey in *Appendix 1—figure 3*, panel D), and consider only the non-pseudoautosomal scaffolds (plotted in orange) for the next step.
5. The non-pseudoautosomal scaffold (InputFile6_Shaematobium_5pr100) coverage displays a unimodal distribution, and the 99th percentile as a value log2(Female) = 4.6
6. When visualizing log2(Female) along the ZW linkage group, it appears no single threshold value cannot finely discriminate between Z-specific and pseudoautosomal scaffolds. So we used the log2(Female) = 4.6 threshold as maximum value for Z-specific regions (*Appendix 1—figure 3*, panel F, in orange), and log2(Female) = 4.4 threshold as minimum value for pseudoautosomal regions (in grey). Scaffolds with log2(Female) between these two values were classified as 'ambiguous'.

We then applied this threshold value to all the genome. Used R script is: Script8_Genomics_haematobium (Part II); and corresponding input file is: InputFile5-2_Shaematobium_SoapCov_ForR.txt.

The final 'Z', 'Autosome' or 'Ambiguous' assignment is shown in *Figure 1—source data 3*.

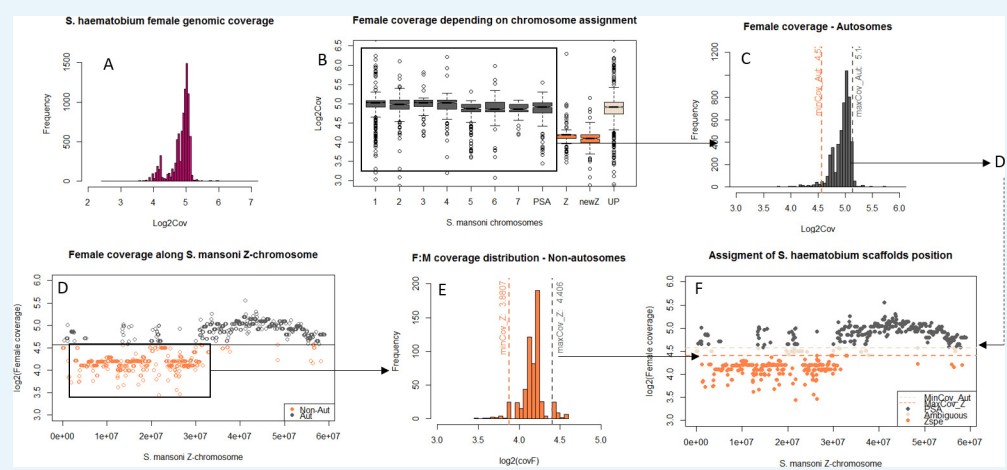

**Appendix 1—figure 3** Determination of Z-specific maximum coverage threshold for S. haematobium species.
DOI: https://doi.org/10.7554/eLife.35684.050

## Definition of evolutionary strata

Genes that were Z-specific in both *S. mansoni* and *S. japonicum* were assigned to stratum S0. Genes that were Z-specific only in *S. mansoni* were assigned to stratum S1mans, and genes that were Z-specific in *S. japonicum* were assigned to stratum S1jap.

This assignment was performed separately for two classifications. The exhaustive classification is defined by the comparison of 'Z' or 'Autosome' assignment, based on coverage (see 2.4–2.6), between *S. mansoni* and *S. japonicum* species. The stringent classification is a subset of the exhaustive classification: it only contains genes localized on the ZW linkage map in *S. mansoni* and with a best location on the Z or PSA region in *S. japonicum* (see *Figure 1—source data 1*).

## Transcriptomics

List of RNA-seq libraries used:

Species | Accession number | Library name
S. mansoni | ERR506076 | Mans_F_Ad_1
S. mansoni | ERR506083 | Mans_F_Ad_2
S. mansoni | ERR506084 | Mans_F_Ad_3
S. mansoni | ERR506088 | Mans_M_Ad_1
S. mansoni | ERR506082 | Mans_M_Ad_2
S. mansoni | ERR506090 | Mans_M_Ad_3
S. mansoni | SRR3223444 | Mans_F_Som_2
S. mansoni | SRR3223443 | Mans_F_Som_1
S. mansoni | SRR3223429 | Mans_M_Som_2
S. mansoni | SRR3223428 | Mans_M_Som_1
S. japonicum | SRR4267990 | Jap_F_Som_1
S. japonicum | SRR4279491 | Jap_F_Som_2
S. japonicum | SRR4279833 | Jap_F_Som_3
S. japonicum | SRR4267991 | Jap_M_Som_1
S. japonicum | SRR4279496 | Jap_M_Som_2
S. japonicum | SRR4279840 | Jap_M_Som_3
S. japonicum | SRR4296940 | Jap_F_Ad_1
S. japonicum | SRR4296942 | Jap_F_Ad_2
S. japonicum | SRR4296944 | Jap_F_Ad_3
S. japonicum | SRR4296941 | Jap_M_Ad_1
S. japonicum | SRR4296943 | Jap_M_Ad_2
S. japonicum | SRR4296945 | Jap_M_Ad_3

### Raw read processing

Reads were first trimmed with Trimmomatic (v0.36, [*Bolger et al., 2014*]), using the following commands:

(i) *S. mansoni*

```
READS =~ pathwaytoreads

cd ${READS}

for i in 'ls | grep. fastq'; do
echo ${i}
 module load java
java -jar ~/tools/Trimmomatic-0.36/trimmomatic-0.36.jar PE -phred33 ${i}
  ${j}.trimmed.fq.gz   ILLUMINACLIP:~/tools/Trimmomatic-0.36/adapters/Tru-
Seq3-SE.fa:2:30:10
```

```
HEADCROP:12 LEADING:3 TRAILING:3 SLIDINGWINDOW:4:15 MINLEN:36
done
```

   (ii) *S. japonicum*

```
READS =~ pathwaytoreads

cd ${READS}

for i in 'ls | grep. fastq'; do
echo ${i}
module load java
 java -jar ~/tools/Trimmomatic-0.36/trimmomatic-0.36.jar SE -phred33 ${i}
   ${j}.trimmed.fq.gz   ILLUMINACLIP:~/tools/Trimmomatic-0.36/adapters/Tru-
Seq3-SE.fa:2:30:10
HEADCROP:12 LEADING:3 TRAILING:3 SLIDINGWINDOW:4:15 MINLEN:36
done
```

## Read mapping on reference genome

Reads were mapped to their respective reference genomes with TopHat2 (*83*), an alignment program for RNA-seq reads which takes splice junctions into account.

   (i) *S. mansoni*

```
GENOME =~ pathwaytogenome/schistosoma_mansoni.PRJEA36577.WBPS9.genomic.fa
READS =~ pathwaytoreads
OUF=~/TophatOutput

# Build the indexed genome for Bowtie2

~/tools/bowtie2-2.2.9/bowtie2-build ${genome} refgenome

# Map the RNAseq reads against the reference genome with TopHat2

cd ${READS}
for i in 'ls | grep trimmed.fq'; do
echo ${i}

~tools/tophat-2.1.1.Linux_x86_64/tophat -p 3 --library-type fr-firststrand
--microexon-search -i 10 -I 40000 --min-segment-intron 10 --max-segment-
intron 40000 g 1 -o ${OUF}~pathwaytogenome/refgenome ${READS}/${i}

done
```

   (ii) *S. japonicum*

```
GENOME =~ pathwaytogenome/schistosoma_japonicum.PRJEA34885.WBPS9.genomic.
fa
READS =~ pathwaytoreads
OUF=~/TophatOutput

~tools/tophat-2.1.1.Linux_x86_64/tophat -p 3 --microexon-search -i 10 -I
```

```
40000 -
--min-segment-intron 10 --max-segment-intron 40000 g 1 -o ${OUF}~pathwaytoge-
nome/refgenome
${READS}/${i}
```

## Read count

Reads counts for each gene were obtained from the TopHat2 alignments using HTseq (**84**).

(i) *S. mansoni*

```
INF=~/TopHat_Output
GENOME =~ pathwaytogenome/schistosoma_mansoni.PRJEA36577.WBPS9.genomic.fa
OUF=~/HTseq_Output

cd ${INF}

for i in 'ls | grep accepted_hits.bam'; do
echo ${i}
module load anaconda

htseq-count -f bam -s reverse -m union --idattr gene_id -o ${OUF}/${i}_Rev.
htseq ${i}
${GENOME} > ${OUF}/${i}_Rev.count
 done
```

(ii) *S. japonicum*

```
INF=~/TopHat_Output
GENOME=~/pathwaytogenome/schistosoma_japonicum.PRJEA34885.WBPS9.genomic.
fa
OUF=~/HTseq_Output

cd ${INF}

for i in 'ls | grep accepted_hits.bam'; do
echo ${i}
module load anaconda

htseq-count -f bam -s no -m union --idattr gene_id -o ${OUF}/${i}_Rev.htseq
${i}
 ${GENOME} > ${OUF}/${i}_Rev.count
done
```

## Alternative pipeline: Kallisto (example shown for *S. mansoni*)

In order to check that our results held independent of the pipeline used to infer expression levels, we further estimated TPM values using Kallisto (**85**).

```
# Create index

INF =~ pathwaytogenome
```

```
CDS=${inf}/schistosoma_mansoni.PRJEA36577.WBPS9.CDS_transcripts.fa  # or
input for S.
   japonicum   would   be:schistosoma_japonicum.PRJEA34885.WBPS9.CDS_tran-
scripts.fa
OUF=~/Kallistooutput

cd ${OUF}

~/tools/kallisto_linux-v0.43.1/kallisto index -i transcripts.idx ${CDS}

# Run Kallisto

READS =~ pathwaytoreads
OUF=~/Kallistooutput
LIST = Input text file containing the library_ID and located in OUF

cd ${READS}

while read i; do
mkdir ${OUF}/${i}

~/tools/kallisto_linux-v0.43.1/kallisto quant -t 16 -i ${OUF}/transcripts.
idx -o
  ${OUF}/${i}  -b  100  --fr-stranded  ${READS}/${i}_forward  ${READS}/${i}
_reverse

done < ${LIST}
```

## Comparative analysis of gene expression

We perform all the expression analyses in R (Script9_Transcriptomics) with the input files (i) InputFile7_Transcriptomics1 corresponding to the newly identified one-to-one orthologs (see 1.2); and (ii) InputFile10_Transcriptomics2 corresponding to the list obtained on WormBase parasite.

In adults, a correlation analysis revealed an inconsistency between replicates, corresponding to the *S. mansoni* female and male adults: ERR506076 and ERR506082. The corresponding heatmaps are shown in *Appendix 1—figure 4*. These two libraries were excluded from further analyses.

Tables resulting from the Loess normalization are InputFile8_SuperTableNormalized_Transcriptomics1_RPKM, InputFile8-2_SuperTableNormalized_LoessOnAll_Transcriptomics1_RPKM, InputFile9_SuperTableNormalized_Transcriptomics1_TPM, and InputFile11_SuperTableNormalized_Transcriptomics2_RPKM.

They can be used directly in the script Script9_Transcriptomics at the step C.

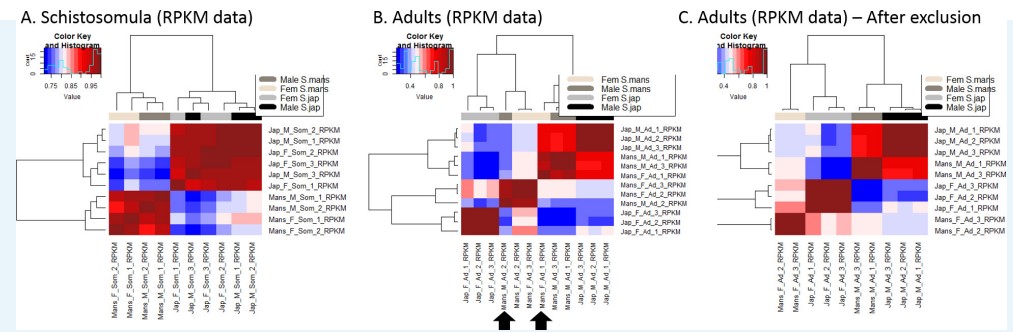

**Appendix 1—figure 4.** Correlation analysis between RNAseq libraries: heatmaps.
DOI: https://doi.org/10.7554/eLife.35684.051

## Number of Protein-Protein interactions (PPIs)

The full set of known PPIs for *S. mansoni* was downloaded from StringDB:
https://stringdb-static.org/download/protein.links.full.v10.5/6183.protein.links.full.v10.5.txt.gz

We kept only interactions supported by experimental evidence or text mining:

```
cat 6183.protein.links.full.v10.5.txt | awk '($10 > 0 || $14 > 0)' | awk '{print
$1, $2}' | perl -pi -e 's/6183\.//gi' | perl -pi -e 's/__mRNA//gi' | perl -pi -e
's/\.[0--9]*//gi'>6183.protein.links.full.v10.5_exp.txt
```

We cleaned the coverage location, so that 'Not analysed (Scaffold <40 Kb)' was replaced by NA:

```
cat InputFile14_PPIs_woNA | perl -pi -e 's/Not analysed \(Scaffold \<40 kb\)/
NA/gi' | sort >InputFile14_PPIs
```

Then counted the number of interactions present for each gene:

```
perl  Script11_PPIcounter  InputFile14_PPIs.txt  6183.protein.links.full.
v10.5_exp.txt
```

Output: five last columns of 'InputFile8-2_SuperTableNormalized_LoessOnAll_Transcriptomics1_RPKM' ($Autocount_exp $Zcount_exp $Excludedcount_exp $NAcount_exp $TotalPPIs_exp)

## Proteomics *vs* microarrays correlation

We performed all the expression analyses in R (Script10_Proteomics_Microarrays) with the input files (i) InputFile12_Proteomics1_Microarrays for which a low value was imputed to undetected proteins*; and (ii) InputFile13_Proteomics2_Microarrays corresponding to the raw data, without imputation.

*such imputation is necessary for the statistical analysis of gene expression differences between males and females (*Figure 4-source data 1* and *2*).

## Obtaining Ka and Ks values for *Schistosoma*

CDS sequences were downloaded from Parasite Wormbase, and 1:1 orthologs detected with a stringent reciprocal best hit approach (Blat with a translated query and database, and a minimum mapping score of 100). These orthologs were aligned with TranslatorX (http://translatorx.co.uk/) with the Gblocks option, and Ka and Ks values were obtained with KaKs_calculator (https://sourceforge.net/projects/kakscalculator2/, model NG). The specific commands used are provided below. Tables of Ka and Ks values are provided as input files 15 and 16.

### S. mansoni/S. haematobium

```
# Get CDS

wget

ftp://ftp.ebi.ac.uk/pub/databases/wormbase/parasite/releases/WBPS10/spe-
cies/schistosoma_mansoni/PRJEA36577/schistosoma_mansoni.PRJEA36577.
WBPS10.CDS_transcripts.fa.gz

gzip -d schistosoma_mansoni.PRJEA36577.WBPS10.CDS_transcripts.fa.gz

mv schistosoma_mansoni.PRJEA36577.WBPS10.CDS_transcripts.fa sman_cds.fa

wget

ftp://ftp.ebi.ac.uk/pub/databases/wormbase/parasite/releases/WBPS10/spe-
cies/schistosoma_haematobium/PRJNA78265/schistosoma_haematobium.
PRJNA78265.WBPS10.CDS_transcripts.fa.gz

gzip -d schistosoma_haematobium.PRJNA78265.WBPS10.CDS_transcripts.fa.gz

mv schistosoma_haematobium.PRJNA78265.WBPS10.CDS_transcripts.fa shaem_cds.
fa

# Run KaKS pipeline

perl  Script12_KaKs.pl  Sman_Shaem/shaem_cds.fa  Sman_Shaem/sman_cds.fa
Sman_Shaem/
```

### S. mansoni/S. japonicum

```
# Get CDS

wget ftp://ftp.ebi.ac.uk/pub/databases/wormbase/parasite/releases/WBPS10/
species/schistosoma_mansoni/PRJEA36577/schistosoma_mansoni.PRJEA36577.
WBPS10.CDS_transcripts.fa.gz

gzip -d schistosoma_mansoni.PRJEA36577.WBPS10.CDS_transcripts.fa.gz

wget

ftp://ftp.ebi.ac.uk/pub/databases/wormbase/parasite/releases/WBPS10/
```

```
species/schistosoma_japonicum/PRJEA34885/schistosoma_japonicum.
PRJEA34885.WBPS10.CDS_transcripts.fa.gz

gzip -d schistosoma_japonicum.PRJEA34885.WBPS10.CDS_transcripts.fa.gz

mv schistosoma_japonicum.PRJEA34885.WBPS10.CDS_transcripts.fa sjap_cds.fa
mv schistosoma_mansoni.PRJEA36577.WBPS10.CDS_transcripts.fa sman_cds.fa

# Run KaKS pipeline

perl    Script12_KaKs.pl    Sman_Sjap/sjap_cds.fa    Sman_Sjap/sman_cds.fa
Sman_Sjap/
```

