## [Decision Letter]

Thank you for submitting your article "Evolution of gene dosage on the Z-chromosome of schistosome parasites: a snapshot of Ohno's hypothesis" for consideration by *eLife*. Your article has been reviewed by three peer reviewers, and the evaluation has been overseen by a Reviewing Editor and Patricia Wittkopp as the Senior Editor. The reviewers have opted to remain anonymous.

The reviewers have discussed the reviews with one another and the Reviewing Editor has drafted this decision to help you prepare a revised submission.

Vicoso and colleagues analyse dosage compensation status of the Z chromosome in ZW females and ZZ males in three schistosome species and propose that the Z-linked genes are upregulated in both sexes in addition to reduced expression of Z-linked genes in females indicating that their results are consistent with Ohno's original hypothesis. Reviewers agreed that studying evolution of sex chromosome is an important topic and that authors provide interesting evidence for dosage compensation regulation in schistosome. However, a number of concerns were raised regarding the analysis of the genomewide data set for scoring dosage compensation status. These points need to be addressed in a thoroughly revised version before we make a final decision on the manuscript.

We would like you to especially address the following concerns.

Reviewers discussed that authors should pay close attention to, and describe in detail, how they calculate X:A ratios. They should take into consideration the expression of house-keeping genes and sex bias expression. This is an important point to address as it could significantly change the interpretations regarding the status of dosage compensation in schistosomes.

This primary concern is described in more detail in the reviewers' original comments, so their full reviews are appended below.

Reviewer #1:

Here, Picard et al. examine the gene content and dosage compensation status of the Z chromosome in ZW female and ZZ male African and Asian schistosomes. Dosage compensation is assayed by comparing Z expression between the sexes (F:M ratio) and by calculating Z versus autosomal expression within each sex (Z:AA ratio). Both ratios are expected to = 1.0 if dosage compensation is "complete". They find that the F:M ratio falls below 1.0, suggesting greater Z expression in males than females. Z:AA ratios in females fall between 0.5 and 1.0, while in males they exceed 1.0. The conclusion is that some degree of Z-upregulation, or "incomplete" dosage compensation, occurs in both sexes, consistent with the first evolutionary step as proposed by Ohno. Analysis of microarray and proteomic data reveals concordance between RNA and protein levels, suggesting that post-transcriptional mechanisms do not kick in to provide "full" Z dosage compensation.

This manuscript addresses an important biological question. However, I'm not sure the analyses currently support the proposed Z dosage compensation status in schistosomes.

Studying Z/X upregulation is complex, and several previous studies are flawed because they have not paid attention to several important factors. In the current submission the authors have considered most of these factors: they have set an appropriate lower cut-off for gene expression (RPKM >1), and have compared expression of extant with ancestral Z-genes. However, the issue of dosage-sensitive housekeeping genes is not considered. As proposed by Birchler and demonstrated by Pessia (cited in the first paragraph of the subsection “The relevance of the Ohno’s hypothesis in the high-throughput sequencing era”), upregulation preferentially affects this specific class of genes. Pessia et al. showed that when all expressed genes are included in X:AA calculations, X:AA ratios in XY males fall between 0.5 and 1.0, suggesting that X-upregulation is "incomplete". However, when only housekeeping genes are considered, X:AA ratios equal 1.0, showing that X-upregulation is actually "complete". A similar approach by Sangrithi et al. (2017) (not cited here) validated and refined this approach, identifying housekeeping genes by virtue of ubiquitous expression. Thus, the status of upregulation will vary depending on which genes are assayed. Also related to this point is whether the authors are studying the same expressed genes across different sexes and tissues. This is the most appropriate comparison.

To address the effect of post-transcriptional regulation on Z dosage compensation, the authors then compare microarray and proteomics data on schistosome heads and gonads. It's not clear to me why the tissues used for this analysis are distinct from those in the RNA-seq analysis; this makes the manuscript a bit awkward. However, setting this aside, the issue of gene choice once again becomes important. Many studies have shown that the mammalian X chromosome is enriched in genes with CNS functions and gonadal functions; the presence of these highly expressed genes artificially elevates X:AA ratios. Is there evidence for such gonadal and neural specialisation of schistosome Z chromosomes? Related to this point, the authors should address whether Z:AA ratios are influenced by imposition of upper expression cut-offs. Such cut-offs affect X:AA ratios in the CNS and gonads (see Sangrithi et al. and studies from Disteche).

Reviewer #2:

The authors use depth of coverage from whole genome sequencing to identify Z-linked genes in three schistosome species (one from Asia and two from Africa). Although the Z chromosome is homologous across all three species, there are differences in the evolutionary strata between species from different continents, suggesting that the lineages differ is some pseudoautosomal regions. When looking at gene expression, several patterns emerge: (i) the Z has lower expression than the autosomes in females, (ii) the Z has higher expression than the autosomes in males, (iii) the Z has strongly male-biased expression. The authors interpret these results as being in agreement with the early stage of Ohno's classic model for the evolution of dosage compensation in which the Z is upregulated in both sexes in order to compensate for its reduced dosage in females. Similar patterns are seen at the level of protein abundance, suggesting that post-transcriptional regulation does not play a major role in the evolution of dosage compensation in these species.

This paper is very well-written and the figures do a good job of illustrating the main results. The authors have made a convincing case and have gone an extra step beyond most previous studies of this type by including proteomic data. I think that this work makes an important contribution to the field. I don't see any major flaws.

Reviewer #3:

The genomics of sex chromosome function and evolution has been under intense study for the last 15 years. These studies, along with work that augments the genetics of dosage compensation, have enriched our understanding of selective pressures and evolution of sex chromosomes. Having additional models is always useful. The current work examines a lineage which allows a comparison of expression sex chromosome linked genes in various stages of being made hemizygous in a ZW system. As the authors note, X0/XY and ZW systems appear to follow different rules for sex chromosomes dosage compensation. The current work confirms these differences in a well written, but under-detailed manuscript.

1) The ortholog mapping and projection of scaffolds onto the Z chromosomes appears robust. This will provide a valuable map for future use of these species in sex chromosome studies as well as for other uses in the genomic parasitology field. It would be useful if these mappings were available. I understand that the TPA has rigorous steps for getting such information into a public repository, but maybe there are other options, such as adding a table in the supplement.

2) There are many confounding factors in using median expression in females and males to measure dosage compensation, none of which appear to be taken into consideration. The sexes of schistosoma are highly dimorphic and are enriched in gonad compared to most organisms that have been examined for sex chromosome dosage compensation. This creates analytical complexities. For example, extensive sex-biased expression, differences in the distribution of genes showing sex-biased expression, differences in cell-type composition between the sexes, or MSCI could all complicate the analysis. Looking at the sexually undifferentiated stages may take care of some of these concerns, but the authors do not explain any of the complexities and workarounds to the reader, whom might not be well acquainted with the field. I would very much like to see the sex-bias profiles, some attempt to examine the expression of more housekeeping genes, and a generally deeper look at the data.

3) That there is not a highly robust translational compensation response is a nice addition to the literature. However, it is important to demonstrate that the proteomics has the sensitivity to make any claims about the contribution of post-transcriptional control to dosage compensation (or lack thereof). The correlations with expression data are positive, but it seems imprudent to extend too far from the primary data showing that F/M ratio for proteins encoded by Z-linked genes is reduced.

---

## [Author Response]

[…] Reviewers discussed that authors should pay close attention to, and describe in detail, how they calculate X:A ratios. They should take into consideration the expression of house-keeping genes and sex bias expression. This is an important point to address as it could significantly change the interpretations regarding the status of dosage compensation in schistosomes.

We agree that taking into account sex-biased genes, and verifying that the results hold for house-keeping genes, are important for the interpretation of the data. We have therefore performed several additional analyses:

1) Sex-biased expression: We have checked that patterns hold when genes with a sex-bias larger than two-fold were excluded (now in Figure 2—figure supplement 15 and 16 and Figure 3—figure supplement 1 and 2I-L panels; and values in Author response tables 1 and 2 below). While the reduced number of genes often leads to a loss of significance within each sex, the general results hold whether or not we consider the ancestral level of expression: (i) a highly significant (***P-value < 0.0001) male-biased expression of the Z, with (F:M)Z/(F:M)A ratio from 0.52 to 0.77, is always observed. (ii) the female Z:AA ratio ranges from 0.69 to 0.93 (sometimes but not always significant). (iii) an over-expression of the Z in males compared to autosomes with ZZ:AA ratio from 1.14 to 1.55, not always significant.

**Author response table 1. resptable1:** Expression values excluding sex-bias FC>2 (ancestry not considered).

	*S. japonicum* schistosomula	*S. japonicum* adults	*S. mansoni* schistosomula	*S. mansoni* adults
Female Z:A	0.86 N.S.	0.81 N.S.	0.93 N.S.	0.69 *
Male Z:A	1.55 ***	1.18 N.S.	1.21*	1.14 N.S.
Z(F:M) / A(F:M)	0.58 ***	0.68 ***	0.67 ***	0.63 ***
Z(fem): Z(male)	0.61 **	0.76 *	0.70 *	0.73 *
A(fem): A(male)	1.04 N.S.	1.11 *	1.05 N.S.	1.15 *
N (Z)	237	189	298	224
N (A)	2,364	1,447	2,364	1,447

**Author response table 2. resptable2:** Expression values excluding sex-bias FC>2 (considering ancestry).

	*S. japonicum* schistosomula	*S. japonicum* adults	*S. mansoni* schistosomula	*S. mansoni* adults
Female Z:A	0.83*	0.71*	0.92 N.S.	0.72 N.S.
Male Z:A	1.23 N.S.	1.38 N.S.	1.20*	1.34 N.S.
Z(F:M) / A(F:M)	0.67 ***	0.52***	O.77***	0.54***
Z(fem): Z(male)	0.65*	0.50*	0.78 N.S.	0.56*
A(fem): A(male)	0.98 N.S.	0.97 N.S.	1.02 N.S.	1.03 N.S.
N (Z)	50	31	111	66
N (A)	2,364	1,447	2,364	1,447

2) House-keeping genes: Since house-keeping genes are not as well defined in schistosomes as in model organisms, we repeated our analysis with genes that had significant levels of expression (RPKM>1 and RPKM>3) in both sexes and both stages. These should have at least very broad expression, and be strongly enriched for house-keeping genes. Again, the results were very similar to those obtained for the full gene set: Z-linked genes were significantly and strongly male-biased in every case, and a significant increase in male expression of the Z was detected in most cases (Figure 2—figure supplement 15 and 16 and Figure 3—figure supplement 1 and 2A-G).

3) Dosage sensitive genes: We enriched our sample for dosage-sensitive genes by considering only genes with known protein-protein interactions (obtained from StringDB, and keeping only those with experimental or text-mining support). While this left us with a very limited set of genes (32 Z-specific genes in *S. mansoni* and 30 in *S. japonicum*), the male-bias in expression of Z-specific genes was always apparent and significant (paragraph 3.6 in Appendix 1 and Figure 2—figure supplement 17), again supporting a very general pattern rather than a bias that affects only genes with sex-specific functions.

This primary concern is described in more detail in the reviewers' original comments, so their full reviews are appended below.

Reviewer #1:

[…] This manuscript addresses an important biological question. However, I'm not sure the analyses currently support the proposed Z dosage compensation status in schistosomes.

We appreciate the enthusiasm for the questions at hand, and have performed several additional analyses to address these concerns (see also our response to the editorial summary above).

Studying Z/X upregulation is complex, and several previous studies are flawed because they have not paid attention to several important factors. In the current submission the authors have considered most of these factors: they have set an appropriate lower cut-off for gene expression (RPKM >1), and have compared expression of extant with ancestral Z-genes. However, the issue of dosage-sensitive housekeeping genes is not considered. As proposed by Birchler and demonstrated by Pessia (cited in the first paragraph of the subsection “The relevance of the Ohno’s hypothesis in the high-throughput sequencing era”), upregulation preferentially affects this specific class of genes. Pessia et al. showed that when all expressed genes are included in X:AA calculations, X:AA ratios in XY males fall between 0.5 and 1.0, suggesting that X-upregulation is "incomplete". However, when only housekeeping genes are considered, X:AA ratios equal 1.0, showing that X-upregulation is actually "complete". A similar approach by Sangrithi et al. (2017) (not cited here) validated and refined this approach, identifying housekeeping genes by virtue of ubiquitous expression. Thus, the status of upregulation will vary depending on which genes are assayed.

We fully agree, and the novelty of our results does not stem from the incomplete up-regulation of female expression of Z-linked genes, which had been described before. It stems from the fact that, as suggested, the detection of this incomplete upregulation seems to be largely dependent on filtering, and is not sufficient to account for the consistent male-bias of expression observed (which does hold for genes with broad expression and more protein-protein interactions [see our new Figure 2—figure supplement 15 to Figure 3—figure supplement 2]). We have now edited the text to make this clearer, and added the missing citation:

Results, subsection “2. Consistent patterns of expression in *S. mansoni* and *S. japonicum*”: ʺWhile this [the absence of global dosage compensation] was generally supported by the lower expression levels of Z-specific genes in females when compared to the autosomes (Z:AA ratio between 0.73 and 0.85; Figure 2, Supplementary file 1), this difference was only apparent for some filtering procedures (Figure 2—figure supplements 1-14), and even then was not sufficient to fully account for the strong male-bias of the Z."

Results, subsection “3. Convergent upregulation of the Z in both sexes”: ʺFigure 3 confirms that the male-biased expression of Z-specific genes is a consequence of their sex-linkage, and that the Z-chromosome has become under-expressed in females relative to the ancestral expression. However, a full two-fold reduction in female expression is not observed, consistent with partial upregulation, and/or full upregulation of a subset of dosage-sensitive genes (Z:AA ranging from 0.68 to 0.83, Supplementary file 1)."ʺ

Also related to this point is whether the authors are studying the same expressed genes across different sexes and tissues. This is the most appropriate comparison.

While in the main figures genes with RPKM>1 in both sexes of the specific stage under study are used, we have now added supplementary figures with only genes with RPKM>1 in both sexes and stages (Figure 2—figure supplement 15 and 16 without ancestry, and Figure 3—figure supplement 1 and 2 with ancestry). These do not affect our conclusions.

To address the effect of post-transcriptional regulation on Z dosage compensation, the authors then compare microarray and proteomics data on schistosome heads and gonads. It's not clear to me why the tissues used for this analysis are distinct from those in the RNA-seq analysis; this makes the manuscript a bit awkward.

We now justify the choice of these tissues:

Results, subsection “4. Male-biased protein dosage of Z-specific genes”: ʺHeads and gonads were chosen as they allowed us to compare Z-specific gene dosage in tissues with widespread functional sex-specificity (ovary and testis), and in a tissue where most dosage imbalances are likely to be deleterious."

However, setting this aside, the issue of gene choice once again becomes important. Many studies have shown that the mammalian X chromosome is enriched in genes with CNS functions and gonadal functions; the presence of these highly expressed genes artificially elevates X:AA ratios. Is there evidence for such gonadal and neural specialisation of schistosome Z chromosomes

This is not entirely straightforward to address, since if the Z-chromosome generally has increased expression in male tissues, it may seem to carry an ʺexcess" of genes expressed in the testis and in the (male) brain. However: (1) This enrichment should hold for all male tissues, not just brain and testis. (2) There should be a depletion of such highly expressed genes on the Z when female tissues are used instead, if the female Z is generally under-expressed (whereas an excess of brain genes should still be detected in female samples if neural genes were generally over-represented).

We could to some extent test this using the microarray data, for which there were three female and male tissues (gonads, heads, hind; the latter is completely distinct from the two first, since heads and gonads are at the front/middle of the animals). Author response image 1 shows that the Z is indeed enriched for genes highly expressed in all three male tissues (although this is for the most part not significant), and depleted of genes highly expressed in the female tissues (Author response image 2), in agreement with a general over-expression of Z-linked genes in all male tissues.

[Brief summary of how these figures were obtained: we used the same microarray dataset as for the proteomics comparison, to which we added the coverage-based Z/autosome assignments (our ʺexhaustive classificationʺ). Two replicates were available for the testis; these were Loess-normalized using the R package ʺAffy" and averaged. A single replicate was available for the other tissues. We then estimated the percentage of Z-linked and autosomal genes that were among the top 50/80/90-th percentile of expression for each male and female tissue.]

**Author response image 1. respfig1:** Enrichment of the Z-chromosome for genes highly expressed in three male tissues (testis, head, hind), for three different expression cut-offs. *p<0.05 **p<0.01, Pearsonʹs Chi-squared test with Yatesʹ continuity correction.

**Author response image 2. respfig2:** Depletion of the Z-chromosome for genes highly expressed in three female tissues (ovary, head, hind), for three different expression cut-offs. *p<0.05 **p<0.01, Pearson'ʹs Chi-squared test with Yates' continuity correction.

Related to this point, the authors should address whether Z:AA ratios are influenced by imposition of upper expression cut-offs. Such cut-offs affect X:AA ratios in the CNS and gonads (see Sangrithi et al. and studies from Disteche)

The tables below show that applying even a stringent upper expression cut-off (RPKM<5) does not change the results. Since this does not affect our conclusions, we have not added this to our manuscript, but would of course be happy to do so if it is felt necessary.

**Author response table 3. resptable3:** Ratio of expression values, with minimum threshold of RPKM>1 and maximum expression threshold of RPKM<5. We did not account for ancestry. Level of significance (Wilcoxon test): *P-value<0.05, **P-value<0.001, ***P-value<0.0001, N.S. P-value>0.05.

	*S. japonicum* schistosomula	*S. japonicum* adults	*S. mansoni* schistosomula	*S. mansoni* adults
Female Z:A	0.76**	0.67**	0.76***	0.76*
Male Z:A	1.39***	1.22*	1.18*	1.33*
Z(F:M) / A(F:M)	0.57***	0.63***	0.69***	0.64***
Z(fem): Z(male)	0.59***	0.68**	0.73***	0.72**
A(fem): A(male)	1.04	1.07	1.06*	1.12*
N (Z)	43	38	55	46
N (A)	486	321	486	321

**Author response table 4. resptable4:** Ratio of expression values, with minimum expression threshold of RPKM>1 and maximum expression threshold of RPKM<5. We normalized by the ancestral state of expression. Level of significance (Wilcoxon test *P-value<0.05, **P-value<0.001, ***P-value<0.0001, N.S. P-value>0.05.

	*S. japonicum* schistosomula	*S. japonicum* adults	*S. mansoni* schistosomula	*S. mansoni* adults
Female Z:A	0.82*	0.79 N.S.	0.90 N.S.	1.05 N.S.
Male Z:A	1.22 N.S.	1.17 N.S.	1.22 N.S.	1.50*
Z(F:M) / A(F:M)	0.67***	0.67*	0.74*	0.70*
Z(fem): Z(male)	0.62*	0.66 N.S.	0.79 N.S.	0.72 N.S.
A(fem): A(male)	0.93 N.S.	0.98 N.S.	1.07 N.S.	1.02 N.S.
N (Z)	11	8	23	16
N (A)	486	321	486	321

Reviewer #3:

The genomics of sex chromosome function and evolution has been under intense study for the last 15 years. These studies, along with work that augments the genetics of dosage compensation, have enriched our understanding of selective pressures and evolution of sex chromosomes. Having additional models is always useful. The current work examines a lineage which allows a comparison of expression sex chromosome linked genes in various stages of being made hemizygous in a ZW system. As the authors note, X0/XY and ZW systems appear to follow different rules for sex chromosomes dosage compensation. The current work confirms these differences in a well written, but under-detailed manuscript.

We appreciate the encouraging comments. We tried to keep the manuscript easy to follow by putting some of the more technical details of our pipeline in Appendix 1, which we hope is sufficiently detailed to allow other researchers to easily reproduce our analysis. However, we will of course be happy to follow any specific suggestions about moving information to the main text, if it is felt that this will help readers follow the manuscript.

1) The ortholog mapping and projection of scaffolds onto the Z chromosomes appears robust. This will provide a valuable map for future use of these species in sex chromosome studies as well as for other uses in the genomic parasitology field. It would be useful if these mappings were available. I understand that the TPA has rigorous steps for getting such information into a public repository, but maybe there are other options, such as adding a table in the supplement.

We fully agree that as much of the data as possible should be made available to other researchers. We currently provide all our input files, which include the chromosomal assignment of all orthologs in the three species (Figure 1—source data 1), as well as the assignment for all genomic scaffolds in Figure 1—source data 2, 3 and 4. While for the moment the scripts and text-only versions of the input files that were used for each analysis are in Dropbox folders (listed in Appendix 1), we will also upload their final versions to the public IST Austria Data Repository (https://datarep.app.ist.ac.at/).

2) There are many confounding factors in using median expression in females and males to measure dosage compensation, none of which appear to be taken into consideration. The sexes of schistosoma are highly dimorphic and are enriched in gonad compared to most organisms that have been examined for sex chromosome dosage compensation. This creates analytical complexities. For example, extensive sex-biased expression, differences in the distribution of genes showing sex-biased expression, differences in cell-type composition between the sexes, or MSCI could all complicate the analysis. Looking at the sexually undifferentiated stages may take care of some of these concerns, but the authors do not explain any of the complexities and workarounds to the reader, whom might not be well acquainted with the field. I would very much like to see the sex-bias profiles, some attempt to examine the expression of more housekeeping genes, and a generally deeper look at the data.

As pointed out by the reviewer, we bypass the issue of sex-biased genes (and in particular of gonad expression, which can be affected by MSCI and other biases), by using either a sexually immature stage or a somatic tissue. We now mention this explicitly in the text:

Results, subsection “2. Consistent patterns of expression in *S. mansoni* and *S. japonicum*.“: ʺThe inclusion of a sexually immature stage is important, as much of the expression obtained from adults will necessarily come from their well-developed gonads. Sex-linked genes are often sex-biased in the germline, even in organisms that have chromosome-wide dosage compensation (e.g. due to sex-chromosome inactivation during gametogenesis), and the inclusion of gonad expression has led to inconsistent assessments of the status of dosage compensation in other clades (16,29)."

Results, subsection “4. Male-biased protein dosage of Z-specific genes.”: ʺHeads and gonads were chosen as they allowed us to compare Z-specific gene dosage in tissues with widespread functional sex-specificity (ovary and testis), and in a tissue where most dosage imbalances are likely to be deleterious."

We have also repeated our analysis by: (1) Removing genes with a sex-bias of expression greater than 2-fold, to remove most genes with sex-specific functions; (2) Using only genes with broad expression (RPKM>1 and RPKM>3 in all samples), to enrich our sample for house-keeping genes (now in Figure 2—figure supplement 15 and 16 and Figure 3—Figure supplement 1 and 2). While the numbers change, these filtering procedures do not affect the general conclusions (strong and consistent male-biased expression of the Z in all samples, small reduction in female expression of the Z in most samples, small increase in male expression of the Z in most samples).

3) That there is not a highly robust translational compensation response is a nice addition to the literature. However, it is important to demonstrate that the proteomics has the sensitivity to make any claims about the contribution of post-transcriptional control to dosage compensation (or lack thereof). The correlations with expression data are positive, but it seems imprudent to extend too far from the primary data showing that F/M ratio for proteins encoded by Z-linked genes is reduced.

We have changed the text to avoid making overly strong claims of absence of post-transcriptional mechanisms:

Abstract: "Quantitative proteomics suggest post-transcriptional mechanisms do not play a major role in balancing the expression of Z-linked genes."

We changed the title of Results section 4 to: "4. Male-biased protein dosage of Z-specific genes.”

Discussion, subsection “The relevance of the Ohno’s hypothesis in the high-throughput sequencing era”: ʺOur results, which consider ancestral expression and do not indicate a major influence of post-transcriptional regulation, suggest a scenario closer to Ohno’s original hypothesis, with the male Z showing a consistent increase in expression."